# HyperFace: Generating Synthetic Face Recognition Datasets by Exploring Face Embedding Hypersphere

**Hatef Otroshi Shahreza**[1,2] **and Sébastien Marcel**[1,3]
[1]Idiap Research Institute, Martigny, Switzerland
[2]École Polytechnique Fédérale de Lausanne (EPFL), Lausanne, Switzerland
[3]Université de Lausanne (UNIL), Lausanne, Switzerland
{hatef.otroshi,sebastien.marcel}@idiap.ch

## Abstract

Face recognition datasets are often collected by crawling Internet and without individuals' consents, raising ethical and privacy concerns. Generating synthetic datasets for training face recognition models has emerged as a promising alternative. However, the generation of synthetic datasets remains challenging as it entails adequate inter-class and intra-class variations. While advances in generative models have made it easier to increase intra-class variations in face datasets (such as pose, illumination, etc.), generating sufficient inter-class variation is still a difficult task. In this paper, we formulate the dataset generation as a packing problem on the embedding space (represented on a hypersphere) of a face recognition model and propose a new synthetic dataset generation approach, called HyperFace. We formalize our packing problem as an optimization problem and solve it with a gradient descent-based approach. Then, we use a conditional face generator model to synthesize face images from the optimized embeddings. We use our generated datasets to train face recognition models and evaluate the trained models on several benchmarking real datasets. Our experimental results show that models trained with HyperFace achieve state-of-the-art performance in training face recognition using synthetic datasets. Project page: https://www.idiap.ch/paper/hyperface

## 1 Introduction

Recent advances in the development of face recognition models are mainly driven by the deep neural networks (He et al., 2016), the angular loss functions (Deng et al., 2019; Kim et al., 2022), and the availability of large-scale training datasets (Guo et al., 2016; Cao et al., 2018; Zhu et al., 2021). The large-scale training face recognition datasets are collected by crawling the Internet and without the individual's consent, raising privacy concerns. This has created important ethical and legal challenges regarding the collecting, distribution, and use of such large-scale datasets (Nat, 2022). Considering such concerns, some popular face recognition datasets, such as MS-Celeb (Guo et al., 2016) and VGGFace2 (Cao et al., 2018), have been retracted.

With the development of generative models, generating synthetic datasets has become a promising solution to address privacy concerns in large-scale datasets (Melzi et al., 2024; Shahreza et al., 2024). In spite of several face generator models in the literature (Deng et al., 2020; Karras et al., 2019; 2020; Rombach et al., 2022; Chan et al., 2022), generating a synthetic face recognition model that can replace real face recognition datasets and be used to train a new face recognition model from scratch is a challenging task. In particular, the generated synthetic face recognition datasets require adequate *inter-class* and *intra-class* variations. While conditioning the generator models on different attributes can help increasing *intra-class* variations, increasing *inter-class* variations remains a difficult task.

In this paper, we focus on the generation of synthetic face recognition datasets and formulate the dataset generation process as a packing problem on the embedding space (represented on the surface

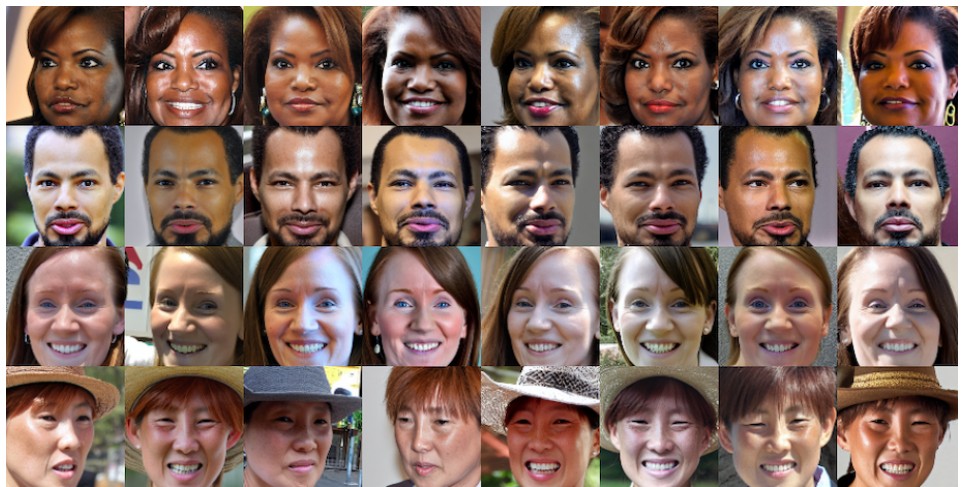

Figure 1: Sample face images from the HyperFace dataset

of a hypersphere) of a pretrained face recognition model. We investigate different packing strategies and show that with a simple optimization, we can find a set of reference embeddings for synthetic subjects that has a high inter-class variation. We also propose a regularization term in our optimization to keep the optimized embedding on the manifold of face embeddings. After finding optimized embeddings, we use a face generative model that can generate face images from embeddings on the hypersphere, and generate synthetic face recognition datasets. We use our generated synthetic face recognition datasets, called HyperFace, to train face recognition models. We evaluate the recognition performance of models trained using synthetic datasets, and show that our optimization and packing approach can lead to new synthetic datasets that can be used to train face recognition models. We also compare trained models with our generated dataset to models trained with previous synthetic datasets, where our generated datasets achieve competitive performance with state-of-the-art synthetic datasets in the literature. Figure 1 illustrates sample face images from our synthetic dataset.

The remainder of this paper is organized as follows. In Section 2, we present our problem formulation and describe our proposed method to generate synthetic face datasets. In Section 3, we provide our experimental results and evaluate our synthetic datasets. In Section 4, we review related work in the literature. Finally, we conclude the paper in Section 5.

## 2 PROBLEM FORMULATION AND PROPOSED METHOD

### 2.1 PROBLEM FORMULATION

**Identity Hypersphere:** Let us assume that we have a pretrained face recognition model $F : \mathcal{I} \to \mathcal{X}$, which can extract identity features (a.k.a. embedding) $\boldsymbol{x} \in \mathcal{X} \subset \mathbb{R}^{n_{\mathcal{X}}}$ from each face image $\boldsymbol{I} \in \mathcal{I}$. Without loss of generality, we can assume that the extracted identity features cover a unit hypersphere[1], i.e., $||\boldsymbol{x}||_2 = 1, \forall \boldsymbol{x} \in \mathcal{X}$.

**Representing Synthetic Dataset on the Identity Hypersphere:** We can represent a synthetic face recognition dataset $\mathcal{D}$ on this hypersphere by finding the embeddings for each image in the dataset. For simplicity, let us assume that for subject $i$ in the synthetic face dataset, we can have a reference face image $\boldsymbol{I}_{\text{ref},i}$ and reference embedding $\boldsymbol{x}_{\text{ref},i} = F(\boldsymbol{I}_{\text{ref},i})$. Note that the reference face image $\boldsymbol{I}_{\text{ref},i}$ may already exist in the synthetic dataset $\mathcal{D}$, otherwise we can assume the reference embedding $\boldsymbol{x}_{\text{ref},i}$ as the average of embeddings of all images for subject $i$ in the dataset $\mathcal{D}$. Therefore, the synthetic face recognition dataset $\mathcal{D}$ with $n_{\text{id}}$ number of subjects can be represented as a set of reference embeddings $\{\boldsymbol{x}_{\text{ref},i}\}_{i=1}^{n_{\text{id}}}$.

---

[1] If the identity embedding $\boldsymbol{x}$ extracted by $F(.)$ is not normalized, we normalize it such that $||\boldsymbol{x}||_2 = 1$.

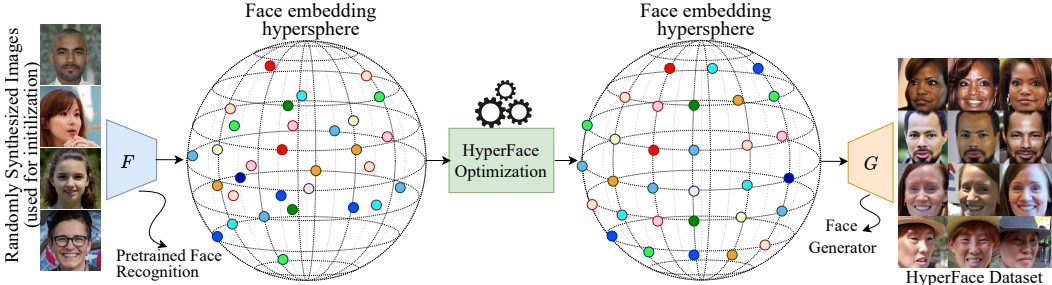

Figure 2: Block diagram of HyperFace Dataset Generation: We start from randomly synthesized face images and extract their embeddings using a pretrained face recognition model $F$. The extracted embeddings are normalised and used as initial points $\{\boldsymbol{x}_{\text{ref},i}\}_{i=1}^{n_{\text{id}}}$ in our HyperFace optimization. The HyperFace optimization tries to increase the *intra-class* variation for synthetic identities on the manifold of the face recognition model over the hypersphere using a regularization term. The resulting points are then used by a face generator model $G$, which can generate synthetic face images from the embeddings.

## 2.2 HYPERFACE SYNTHETIC FACE DATASET

**HyperFace Optimization Problem:** By representing a synthetic dataset $\mathcal{D}$ on the identity hypersphere as a set of reference embeddings $\{\boldsymbol{x}_{\text{ref},i}\}_{i=1}^{n_{\text{id}}}$, we can raise the question that "*How should reference embeddings cover the identity hypersphere?*" To answer this question, we remind that the distances between reference embeddings indicate the inter-class variation in the synthetic face recognition dataset $\mathcal{D}$. Therefore, since we would like to have a high inter-class variation in the generated dataset $\mathcal{D}$, we can say that we need to maximize the distances between reference embeddings $\{\boldsymbol{x}_{\text{ref},i}\}_{i=1}^{n_{\text{id}}}$. In other words, we need to solve the following optimization problem:

$$\max_{\{\boldsymbol{x}_{\text{ref}}\}, i \neq j} \min d(\boldsymbol{x}_{\text{ref},i}, \boldsymbol{x}_{\text{ref},j}) \qquad \text{subject to} \quad ||\boldsymbol{x}_{\text{ref},k}||_2 = 1, \forall k \in \{1, ..., n_{\text{id}}\} \qquad (1)$$

where $d(\cdot, \cdot)$ is a distance function.

**Solving the HyperFace Optimization:** The optimization problem stated in Eq. 1 is a well-known optimization problem, which is known as spherical code optimization (J. H. Conway, 1998) or the Tammes problem (Tammes, 1930), where the goal is to pack a given number of points (e.g., particles, pores, electrons, etc.) on the surface of a unit sphere such that the minimum distance between points is maximized. The optimal solutions for this problem are studied for small dimensions and the number of points (Böröczky, 1983; Hárs, 1986; Musin & Tarasov, 2012; 2015). However, for a large dimension and a high number of points, there is no closed-form solution in the literature (Tokarchuk et al., 2024). For a large dimension and a high number of points, there are different approaches for solving this optimization problem (such as geometric optimization, numerical optimization, etc.). However, for a large dimension of hypersphere (i.e., $n_{\mathcal{X}}$) and a *very* large number of points (i.e., the number of subjects $n_{\text{id}}$, such as 10,000 identities in our problem), solving this optimization can be computationally expensive. To address this issue, we solve the optimization problem with an iterative approach based on gradient descent. To this end, we can randomly initialize the reference embeddings and find the optimised reference embeddings $\{\boldsymbol{x}_{\text{ref},i}\}_{i=1}^{n_{\text{id}}}$ using the Adam optimizer (Kingma & Ba, 2015). This allows us to solve the optimization with a reasonable computation resource. For example, we can solve the optimization for $n_{\mathcal{X}} = 512$ and $n_{\text{id}} = 10,000$ on a system equipped with a single NVIDIA 3090 GPU in 6 hours.

**Regularization:** While we solve the optimization problem in Eq. 1 on the surface of a hypersphere, we should note that the manifold of embeddings $\mathcal{X}$ does not necessarily cover the whole surface of the hypersphere. This means if we get out of the distribution of embeddings $\mathcal{X}$, we may not be able to generate face images from such embeddings. Therefore, we need to add a regularization term to our optimization problem that tends to keep the reference embeddings on the manifold

---

**Algorithm 1** HyperFace Optimization for Finding Reference Embeddings

1: **Inputs**:    $\lambda$ : learning rate, $n_{\text{itr}}$ : number of iterations, $\{\boldsymbol{x}_g\}_{g=1}^{n_{\text{gallery}}}$ : embeddings of a gallery of face images,
2:        $\alpha$ : hyperparameter (contribution of regularization).
3: **Output**:    $\boldsymbol{X}_{\text{ref}} = \{\boldsymbol{x}_{\text{ref},i}\}_{i=1}^{n_{\text{id}}}$ : optimized reference embeddings.
4: **Procedure:**
5:     Initialize reference embeddings $\{\boldsymbol{x}_{\text{ref},i}\}_{i=1}^{n_{\text{id}}}$
6:     **For** $n = 1, .., n_{\text{itr}}$ **do**
7:       Find $\boldsymbol{x}_{\text{ref},i}, \boldsymbol{x}_{\text{ref},j} \in \boldsymbol{X}_{\text{ref}}$ which have minimum distance $d(\boldsymbol{x}_{\text{ref},i}, \boldsymbol{x}_{\text{ref},j})$
8:       Reg $\leftarrow \frac{1}{n_{\text{id}}} \sum_{k=1}^{n_{\text{id}}} \min_{\{\boldsymbol{x}_g\}_{\text{gallery}}} d(\boldsymbol{x}_{\text{ref},k}, \boldsymbol{x}_g)$           ▷ Calculate the regularization term
9:       cost $\leftarrow -d(\boldsymbol{x}_{\text{ref},i}, \boldsymbol{x}_{\text{ref},j})$
10:      $\boldsymbol{X}_{\text{ref}} \leftarrow \boldsymbol{X}_{\text{ref}} - \text{Adam}(\nabla\text{cost}, \lambda)$
11:      $\boldsymbol{X}_{\text{ref}} \leftarrow \text{normalize}(\boldsymbol{X}_{\text{ref}})$     ▷ To ensure that resulting embeddings $\boldsymbol{X}_{\text{ref}}$ remain on the hypersphere.
12:     **End For**
13: **End Procedure**

---

of embeddings $\mathcal{X}$. To this end, we consider a set of face images $\{\boldsymbol{I}_i\}_{i=1}^{n_{\text{gallery}}}$ as a gallery of images[2] and extract their embeddings to have set of valid embeddings $\{\boldsymbol{x}_i\}_{i=1}^{n_{\text{gallery}}}$. Then, we try to minimize the distance of our reference embeddings $\{\boldsymbol{x}_{\text{ref},i}\}_{i=1}^{n_{\text{id}}}$ to the set of embeddings $\{\boldsymbol{x}_i\}_{i=1}^{n_{\text{gallery}}}$, which approximates the manifold of embeddings $\mathcal{X}$. To this end, for each reference embedding $\{\boldsymbol{x}_{\text{ref},i}\}_{i=1}^{n_{\text{id}}}$, we find the closest embedding in $\{\boldsymbol{x}_i\}_{i=1}^{n_{\text{gallery}}}$ and minimize their distance. We can write the optimization in Eq. 1 as a regularized min-max optimization as follows:

$$
\min_{} \quad \max_{\{\boldsymbol{x}_{\text{ref}}\}, i \neq j} -d(\boldsymbol{x}_{\text{ref},i}, \boldsymbol{x}_{\text{ref},j}) + \alpha \underbrace{\frac{1}{n_{\text{id}}} \sum_{k=1}^{n_{\text{id}}} \min_{\{\boldsymbol{x}_g\}_{g=1}^{n_{\text{gallery}}}} d(\boldsymbol{x}_{\text{ref},k}, \boldsymbol{x}_g)}_{\text{regularization}};
$$
$$
\text{subject to} \quad ||\boldsymbol{x}_{\text{ref},k}||_2 = 1, \forall k \in \{1, ..., n_{\text{id}}\}, \tag{2}
$$

where $\alpha$ is a hyperparameter that controls the contribution of the regularization term in the optimization. To provide more flexibility in our optimization, we consider the size of gallery $n_{\text{gallery}}$ to be greater or equal to the number of identities $n_{\text{id}}$ in the synthetic dataset (i.e., $n_{\text{gallery}} \geq n_{\text{id}}$).

**Initialization:**    To solve the HyperFace optimization problem in Eq. 1 using Algorithm 1, we need to initialize the reference embeddings $\{\boldsymbol{x}_{\text{ref},i}\}_{i=1}^{n_{\text{id}}}$. To this end, we can generate $n_{\text{id}}$ number random synthetic images $\{\boldsymbol{I}_i\}_{i=1}^{n_{\text{id}}}$ using a face generator model, such as StyleGAN (Karras et al., 2020), Latent Diffusion Model (LDM) (Rombach et al., 2022). These models use a noise vector as the input and can generate synthetic face images in an unconditional setting. Then, after generating $\{\boldsymbol{I}_i\}_{i=1}^{n_{\text{id}}}$ images, we can extract their embeddings using the face recognition model $F(\cdot)$ and use the extracted embeddings as initialization values for the reference embeddings $\{\boldsymbol{x}_{\text{ref},i}\}_{i=1}^{n_{\text{id}}}$ in Algorithm 1.

**Image Generation:**    After we find the reference embeddings $\{\boldsymbol{x}_{\text{ref},i}\}_{i=1}^{n_{\text{id}}}$ using Algorithm 1, we can use an identity-conditioned image generator model to generate face images from reference embeddings. To this end, we use a recent face generator network (Papantoniou et al., 2024), which is based on probabilistic diffusion models. The diffusion face generator model $G(\cdot, \cdot)$ can generate a face image $\boldsymbol{I} = G(\boldsymbol{x}_{\text{ref}}, \boldsymbol{z})$ from reference embedding $\boldsymbol{x}_{\text{ref}}$ and a random noise $\boldsymbol{z} \sim \mathcal{N}(0, \mathbb{I}^{\text{DM}})$. Therefore, by changing the random noise $\boldsymbol{z}$ and sampling different noise vectors, we can generate different samples for the reference embedding $\boldsymbol{x}_{\text{ref}}$. In addition, to increase intra-class variation, we add Gaussian noise $\boldsymbol{v} \sim \mathcal{N}(0, \mathbb{I}^{n_{\mathcal{X}}})$ to the reference embedding $\boldsymbol{x}_{\text{ref}}$, and then normalize it to locate it on the hypersphere. In summary, we can generate different samples for each reference embedding $\boldsymbol{x}_{\text{ref}}$ by changing $\boldsymbol{z}$ and $\boldsymbol{v}$ noise vectors as follows:

$$
\boldsymbol{I} = G(\frac{\boldsymbol{x}_{\text{ref}} + \beta\boldsymbol{v}}{||\boldsymbol{x}_{\text{ref}} + \beta\boldsymbol{v}||_2}, \boldsymbol{z}), \quad \boldsymbol{v} \sim \mathcal{N}(0, \mathbb{I}^{n_{\mathcal{X}}}), \boldsymbol{z} \sim \mathcal{N}(0, \mathbb{I}^{\text{DM}}), \tag{3}
$$

---

[2]The gallery of face images $\{\boldsymbol{I}_i\}_{i=1}^{n_{\text{gallery}}}$ can be generated using an unconditional face generator network such as StyleGAN (Karras et al., 2020), Latent Diffusion Model (LDM) (Rombach et al., 2022), etc.

Table 1: Comparison of recognition performance of face recognition models trained with different synthetic datasets and a real dataset (i.e., CASIA-WebFace). The performance reported for each dataset is in terms of accuracy and best value for each benchmark is emboldened.

| Dataset name | # IDs | # Images | LFW | CPLFW | CALFW | CFP | AgeDB |
|---|---|---|---|---|---|---|---|
| SynFace (Qiu et al., 2021) | 10'000 | 999'994 | 86.57 | 65.10 | 70.08 | 66.79 | 59.13 |
| SFace (Boutros et al., 2022) | 10'572 | 1'885'877 | 93.65 | 74.90 | 80.97 | 75.36 | 70.32 |
| Syn-Multi-PIE (Colbois et al., 2021) | 10'000 | 180'000 | 78.72 | 60.22 | 61.83 | 60.84 | 54.05 |
| IDnet (Kolf et al., 2023) | 10'577 | 1'057'200 | 84.48 | 68.12 | 71.42 | 68.93 | 62.63 |
| ExFaceGAN (Boutros et al., 2023b) | 10'000 | 599'944 | 85.98 | 66.97 | 70.00 | 66.96 | 57.37 |
| GANDiffFace (Melzi et al., 2023) | 10'080 | 543'893 | 94.35 | 76.15 | 79.90 | 78.99 | 69.82 |
| Langevin-Dispersion (Geissbühler et al., 2024) | 10'000 | 650'000 | 94.38 | 65.75 | 86.03 | 65.51 | 77.30 |
| Langevin-DisCo (Geissbühler et al., 2024) | 10'000 | 650'000 | 97.07 | 76.73 | 89.05 | 79.56 | 83.38 |
| DigiFace-1M (Bae et al., 2023) | 109'999 | 1'219'995 | 90.68 | 72.55 | 73.75 | 79.43 | 68.43 |
| IDiff-Face (Uniform) (Boutros et al., 2023a) | 10'049 | 502'450 | 98.18 | 80.87 | 90.82 | 82.96 | 85.50 |
| IDiff-Face (Two-Stage) (Boutros et al., 2023a) | 10'050 | 502'500 | 98.00 | 77.77 | 88.55 | 82.57 | 82.35 |
| DCFace (Kim et al., 2023) | 10'000 | 500'000 | 98.35 | 83.12 | **91.70** | 88.43 | **89.50** |
| **HyperFace [ours]** | 10'000 | 500'000 | **98.50** | **84.23** | 89.40 | **88.83** | 86.53 |
| CASIA-WebFace (Yi et al., 2014) | 10'572 | 490'623 | 99.42 | 90.02 | 93.43 | 94.97 | 94.32 |

where $\beta$ is a hyperparamter that controls the variations to the reference embedding. Figure 2 depicts the block diagram of our synthetic dataset generation process. Algorithm 3 in Appendix F also present a pseudo-code of dataset generation process.

## 3 EXPERIMENTS

### 3.1 EXPERIMENTAL SETUP

**Dataset Generation:** For solving the HyperFace optimization in Algorithm 1, we use an initial learning rate of $\lambda = 0.01$ and reduce the learning rate by power 0.75 every $5,000$ iterations for a total number of iterations $n_{\text{itr}} = 100,000$. We also consider cosine distance, which is commonly used in face recognition systems for the comparison of face embeddings, as our distance function $d(\cdot, \cdot)$. For the hyperparameters $\alpha$ and $\beta$, we consider default values of $0.5$ and $0.01$, respectively, in our experiments. We also consider the size of gallery to be the same as the number of identities, and explore other cases where $n_{\text{gallery}} > n_{\text{id}}$ in our ablation study. We generate 64 images, by default, per each identity in our generated datasets and explore other numbers of images in our ablation study.

We use ArcFace (Deng et al., 2019) as the pretrained face recognition model $F(\cdot)$ with the embedding dimension of $n_{\mathcal{X}} = 512$ and use a pretrained generator model (Papantoniou et al., 2024) to generate face images from ArcFace embeddings. After generating face images, we align all face images using a pretrained MTCNN (Zhang et al., 2016) face detector model. For our regularization, we randomly generate images with StyleGAN (Karras et al., 2020) as default, and investigate other generator models in our ablation study.

**Evaluation:** To evaluate the generated synthetic datasets, we use each generated datasets as a training dataset for training a face recognition model. To this end, we use the iResNet50 backbone and train the model with AdaFace loss function (Kim et al., 2022) using the Stochastic Gradient Descent (SGD) optimizer with the initial learning rate 0.1 and a weight decay of $5 \times 10^{-4}$ for 30 epochs with the batch size of 256. After training the face recognition model with the synthetic dataset, we benchmark the performance of the trained face recognition models on different benchmarking datasets of real images, including Labeled Faces in the Wild (LFW) (Huang et al., 2008), Cross-age LFW (CA-LFW) (Zheng et al., 2017), CrossPose LFW (CP-LFW) (Zheng & Deng, 2018), Celebrities in Frontal-Profile in the Wild (CFP-FP) (Sengupta et al., 2016), and AgeDB-30 (Moschoglou et al., 2017) datasets. For consistency with prior works, we report recognition accuracy calculated using 10-fold cross-validation for each of benchmarking datasets. The source code of our experiments and generated datasets are publicly available[3].

---

[3]Project page: https://www.idiap.ch/paper/hyperface

## 3.2 ANALYSIS

**Comparison with Previous Synthetic Datasets:** We compare the recognition performance of face recognition models trained with our synthetic dataset and previous synthetic datasets in the literature. We use the published dataset for each method and train all models with the same configuration for different datasets to prevent the effect of other hyperparameters (such as number of epochs, batch size, etc.). For a fair comparison, we consider the versions of datasets with a similar number of identities[4], if there are different datasets available for each method. Table 1 compares the recognition performance of face recognition models trained with different synthetic datasets. As the results in this table show, our method achieves state-of-the-art performance in training face recognition using synthetic data. Figure 1 illustrates sample face images from our synthetic dataset. Figure 4 of appendix also presents more sample images from HyperFace dataset.

**Ablation Study:** In our dataset generation method, there are different hyperparameters which can affect the HyperFace optimization and the generated synthetic datasets. Table 2 reports the ablation study on the number of images generated per each synthetic identity in our experiments. As the results in Table 2 show, increasing the number of images per identity improves the recognition performance of trained face recognition model, but it tends to saturate after 64 images per identity.

Table 2: Ablation study on the effect of number of images

| Image/ID | LFW | CPLFW | CALFW | CFP | AgeDB |
|---|---|---|---|---|---|
| 32 | **98.70** | 84.17 | 88.83 | 88.74 | 86.33 |
| 50 | 98.50 | 84.23 | 89.40 | 88.83 | 86.53 |
| 64 | 98.67 | **84.68** | **89.82** | 89.14 | 87.07 |
| 96 | 98.42 | 84.15 | 89.00 | **89.51** | 87.45 |
| 128 | 98.20 | 83.63 | **89.82** | 89.31 | **87.62** |

Table 3 also compares the number of identities in the generated dataset, including 10k, 20k, and 50k identities. As the results in Table 3 show, increasing the number of identities improves the recognition performance of trained face recognition model on the benchmarking datasets. The re-

Table 3: Ablation study on the effect of number of identities

| # IDs | LFW | CPLFW | CALFW | CFP | AgeDB |
|---|---|---|---|---|---|
| 10k | 98.67 | 84.68 | 89.82 | 89.14 | 87.07 |
| 30k | **98.82** | 85.23 | 91.12 | 91.74 | 89.42 |
| 50k | 98.27 | **85.60** | **91.48** | **92.24** | **90.40** |

sults in this table demonstrates that we can still increase the number of identities and scale our dataset generation without saturating the performance. The main issue for increasing the size of dataset is computation resource, which is discussed in detail in Appendix A. We can also reduce the complexity of our optimization for large number of identities, which is discussed in detail in Appendix B.

Table 4 reports the recognition performance achieved for face recognition model trained with datasets with 10k identity and optimized with different numbers of gallery images. As the results in this table shows, increasing the size of gallery improves the performance of the trained model.

Table 4: Ablation study on the effect of $n_{\text{gallery}}$

| $n_{\text{gallery}}$ | LFW | CPLFW | CALFW | CFP | AgeDB |
|---|---|---|---|---|---|
| 10K | 98.53 | 84.00 | 88.92 | **89.34** | 85.9 |
| 20K | 98.50 | **84.32** | **89.28** | 89.17 | 86.00 |
| 50K | **98.72** | 84.23 | 88.72 | 89.19 | **86.85** |

However, with 10,000 images we can still approximate the manifold of face embeddings on the hypersphere.

As another ablation study, we use different source of images for the gallery set to use in our regularization and solve the HyperFace optimization. We use pretrained StyleGAN (Karras et al., 2020) as a GAN-based generator model and a pretrained latent diffusion model (Rombach et al., 2022) as a diffusion-based generator model. We use these generator models and randomly generate some synthetic face images. In addition, for our ablation study, we consider some real images from BUPT dataset (Wang et al., 2019) as a dataset of real face images.

[4]Only in the dataset used for DigiFace (Bae et al., 2023) there are more identities, because there is only one version available for this dataset, which has a greater number of identities compared to other existing synthetic datasets.

As the results in Table 5 show, optimization with images from Style-GAN and LDM lead to comparable performance for the generated face recognition dataset. However, the real images in the BUPT dataset lead to superior performance. This suggests that the synthesized images cannot completely cover the manifold of embeddings and if we use real images as our gallery it can improve the generated dataset and recognition performance of our face recognition model.

Table 5: Ablation study on the type of data in gallery

| Gallery | LFW | CPLFW | CALFW | CFP | AgeDB |
|---|---|---|---|---|---|
| StyleGAN | 98.67 | 84.68 | 89.82 | 89.14 | 87.07 |
| LDM | 98.65 | 84.35 | 89.17 | 89.09 | 86.35 |
| BUPT | **98.70** | **84.77** | **90.03** | **89.16** | **87.13** |

We also study the effect of hyperparameters $\alpha$ and $\beta$ on the generated face recognition dataset. Table 6 reports the ablation study for the contribution of regularization in our optimization ($\alpha$). As the results in this table shows, the regularization enhances the quality of generated dataset and improves the recognition performance of face recognition model. In fact, our regularization term helps our optimization to keep the points on the manifold of face recognition over the hypersphere, and therefore improves the quality of our synthetic dataset.

Table 6: Ablation study on the effect of $\alpha$

| $\alpha$ | LFW | CPLFW | CALFW | CFP | AgeDB |
|---|---|---|---|---|---|
| 0 | 98.40 | 84.15 | 88.87 | 89.31 | 86.48 |
| 0.50 | **98.67** | 84.68 | **89.82** | 89.14 | **87.07** |
| 0.75 | 98.62 | 84.32 | 89.48 | 89.67 | 86.72 |
| 1.0 | 98.55 | **84.72** | 89.10 | **89.76** | 86.63 |

Similarly, Table 7 reports the ablation study for the effect of noise in data generation and augmentation (i.e., hyperparamter $\beta$ in in Eq. 3). As can be seen, the added noise increases the variation for images of each subject and increases the performance of face recognition models trained with the generated datasets. With a larger value of $\beta$, the generated images for each identity have more variations, which increases the performance of the face recognition model trained with our synthetic dataset.

Table 7: Ablation study on the effect of $\beta$

| $\beta$ | LFW | CPLFW | CALFW | CFP | AgeDB |
|---|---|---|---|---|---|
| 0 | 98.53 | 84.00 | 88.92 | 89.34 | 85.90 |
| 0.005 | 98.67 | 84.68 | 89.82 | 89.14 | 87.07 |
| 0.010 | **98.7** | **84.72** | 90.05 | 89.54 | 88.42 |
| 0.020 | 98.4 | 84.05 | **91.32** | **90.13** | **89.83** |

As another experiment, we consider different backbones and train face recognition models with different number of layers. As the results in Table 8 show, increasing the number of layers improve the recognition performance of trained face recognition model. While this is expected and has been observed for training using large-scale face recognition datasets, it sheds light on more potentials in the generated synthetic datasets.

Table 8: Ablation study on the network structure

| Network | LFW | CPLFW | CALFW | CFP | AgeDB |
|---|---|---|---|---|---|
| ResNet18 | 98.33 | 81.38 | 88.53 | 86.03 | 85.27 |
| ResNet34 | 98.5 | 83.47 | 88.88 | 88.29 | 86.42 |
| ResNet50 | 98.67 | 84.68 | 89.82 | 89.14 | 87.07 |
| ResNet101 | **98.73** | **85.43** | **90.05** | **89.54** | **87.52** |

## 3.3 DISCUSSION

**Scaling Dataset Generation:** To increase the size of the synthetic face recognition dataset, we can increase the number of images per identity and also the number of samples per identity. In our ablation study, we investigated the effect of the number of images (Table 2) and the number of identities (Table 3) on the recognition performance of the face recognition model. However, increasing the size of the dataset requires more computation. Increasing the number of images in the dataset has linear complexity in our image generation step (i.e., $\mathcal{O}(n_{\text{images}})$, where $n_{\text{images}}$ is the number of images in the generated dataset). However, the complexity of solving the HyperFace optimization problem with iterative optimization in Algorithm 1 has quadratic complexity (i.e., $\mathcal{O}(n_{\text{id}}^2)$). Therefore, solving this optimization for a larger number of identities requires much more computation

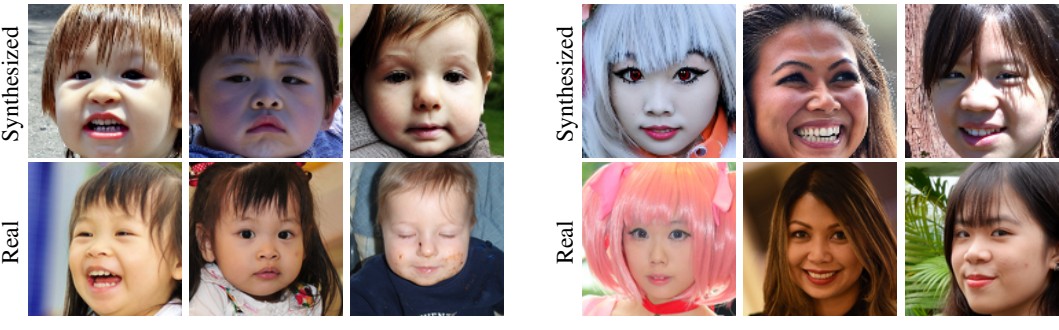

(a) children images                    (b) adult images

Figure 3: Sample pairs of images with the highest similarity between face embeddings of images in synthesized dataset and training dataset of StyleGAN, which was used to generate random images for initialization and regularization in the HyperFace optimization.

resources. Meanwhile, most existing synthetic datasets in the literature have a comparable number of identities to our experiments. We should note that in our optimization, we considered all points in each iteration of optimization which introduces quadratic complexity to our optimization. However, we can solve the optimization with stochastic mini-batches of points on the embedding hypersphere, which can reduce the complexity in each iteration (i.e., $\mathcal{O}(b^2)$, where $b$ is the size of batch and $b \leq n_{\text{id}}$). We further discuss the complexity of our optimization and dataset generation in Appendix A and present further analyses for stochastic optimization, that reduces the complexity of our optimization in Appendix B.

**Leakage of Identity:**    In our dataset generation method, we used images synthesized by StyleGAN for initialization and regularization. Therefore, it is important if there is any leakage of privacy data in the images generated from StyleGAN in the final generated dataset. To this end, we extract and compare embeddings from all the generated images to embeddings of all face images in the training dataset of StyleGAN. The highest similarity score between generated images and training dataset correspond to children images (as shown in Figure 3a) which are difficult to compare visually and conclude potential leakage. Figure 3b illustrates images of highest scores excluding children. While there are some visual similarities in the images, it is difficult to conclude leakage of identity in the generated synthetic dataset. We further study the effect of identity leakage on the recognition performance of face recognition models in Appendix D.

**Ethical Considerations:**    State-of-the-art face recognition models are trained with large-scale face recognition datasets, which are crawled from the Internet, raising ethical and privacy concerns. To address the ethical and privacy concerns with web-crawled data, we can use synthetic data to train face recognition models. However, generating synthetic face recognition datasets also requires face generator models which are trained from a set of real face images. Therefore, we still rely on real face images in the generation pipeline. In our experiments, we investigated if we have leakage of identity in the generated synthetic dataset based on images used for initialization and regularization. However, there are other privacy-sensitive components used in our method. For example, we defined and solved our optimization problem on the embedding hypersphere of a pretrained face recognition model. Therefore, for generating fully privacy-friendly datasets, the leakage of information by other components needs to be investigated.

We should also note that while we tried to increase the inter-class variations in our method, there might be still a potential lack of diversity in different demography groups, stemming from implicit biases of the datasets used for training in our pipeline (such as the pretrained face recognition model, the gallery of images used for regularization, etc.). It is also noteworthy that the project on which the work has been conducted has passed an Institutional Ethical Review Board (IRB).

## 4 RELATED WORK

With the advances in generative models, several synthetic face recognition datasets have been proposed in the literature. Bae et al. (2023) proposed DigiFace dataset where they used a computer-graphic pipeline to render different identities and also generate different images for each identity by introducing different variations based on face attributes (e.g., variation in facial pose, accessories, and textures). In contrast to (Bae et al., 2023) , other papers in the literature used Generative Adversarial Networks (GANs) or probabilistic Diffusion Models (PDMs) to generate synthetic face datasets. Qiu et al. (2021) proposed SynFace and utilised DiscoFaceGAN (Deng et al., 2020) to generate their dataset. They generated different synthetic identities using identity mixup by exploring the latent space of DiscoFaceGAN to increase intra-class variation and then used DiscoFaceGAN to generate different images for each identity.

Boutros et al. (2022) proposed SFace by training an identity-conditioned StyleGAN (Karras et al., 2020) on the CASIA-WebFace (Yi et al., 2014) and then generating the SFace dataset using the trained model. Kolf et al. (2023) also trained an identity-conditioned StyleGAN (Karras et al., 2020) in a three-player GAN framework to integrate the identity information into the generation process and proposed the IDnet dataset. Colbois et al. (2021) proposed the Syn-Multi-PIE dataset using a pretrained StyleGAN (Karras et al., 2020). They trained a support vector machine (SVM) to find directions for different variations (such as pose, illuminations, etc.) in the intermediate latent space of a pretrained StyleGAN. Then, they used StyleGAN to generate different identities and synthesized different images for each identity by exploring the intermediate latent space of StyleGAN using linear combinations of calculated directions. Boutros et al. (2023b) proposed ExFaceGAN, where they used SVM to disentangle the identity information in the latent space of pretrained GANs, and then generated different identities with several images within the corresponding identity boundaries. Geissbühler et al. (2024) used stochastic Brownian forces to sample different identities in the intermediate latent space of pretrained StyleGAN (Karras et al., 2020) and generate different identities (named Langavien). Then they solved a similar dynamical equation in the latent space of StyleGAN to generate different images for each identity (named Langavien-Dispersion) and further explored the intermediate latent space of StyleGAN (named Langavien-DisCo).

Melzi et al. (2023) proposed GANDiffFace, a hybrid dataset generation framework, where they used StyleGAN to generate face images with different identities, and then used DreamBooth (Ruiz et al., 2023) as a diffusion-based generator, to generate different samples for each identity. Boutros et al. (2023a) trained an identity-conditioned diffusion model to generate synthetic face images and proposed IDiffFace datasets. They generated different samples using an unconditional model, and then generated different samples using their conditional diffusion model (named IDiff-Face Two-Stage). Alternatively, they uniformly sampled different identities and generated different samples for each identity using their identity-conditioned diffusion model (named IDiff-Face Uniform). Kim et al. (2023) proposed DCFace, where they trained a dual condition (style and identity conditions) face generator model based on diffusion models on the CASIA-WebFace dataset. They used their trained diffusion model to generate different identities and different styles for each identity by varying identity and style conditions.

## 5 CONCLUSION

In this paper, we formalized the dataset generation as a packing problem on the hypersphere of a pretrained face recognition model. We focused on inter-class variation and designed our packing problem to increase the distance between synthetic identities. Then, we considered our packing problem as a regularized optimization and solved it with an iterative gradient-descent-based approach. Since the manifold of face embeddings does not cover the whole hypersphere, the regularization allows us to approximate the manifold of face embeddings and enhance the quality of generated face images. We used the generated datasets by our method (called HyperFace) to train face recognition models, and evaluated the trained models on several real benchmarking datasets. Our experiments demonstrate the effectiveness of our approach, which achieves state-of-the-art performance for training face recognition using synthetic data. We also presented an extensive ablation study to investigate the effect of each hyperparameter in our dataset generation method.

ACKNOWLEDGMENTS

This research is based upon work supported by the Hasler foundation through the "Responsible Face Recognition" (SAFER) project.

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

## A    COMPLEXITY AND REQUIRED COMPUTATION RESOURCE

The computation required to generate the synthetic datasets in our approach has two main parts:

1. **HyperFace Optimization:** The HyperFace optimization (Algorithm 1) considers all reference points $\{x_{\text{ref},i}\}_{i=1}^{n_{\text{id}}}$ in the hypersphere and maximizes their distances. Therefore, this optimization considers all pairs of points and has quadratic complexity (i.e., $\mathcal{O}(n_{\text{id}}^2)$).

   Table 9 reports the runtime for solving the HyperFace optimization for different numbers of identities (i.e., $n_{\text{id}}$) on a system equipped with a single NVIDIA 3090 GPU. Note that this optimization process cannot be parallelized.

Table 9: Runtime for solving the HyperFace optimization (Algorithm 1) for different numbers of identities on a system equipped with a single NVIDIA 3090 GPU.

| $n_{\text{id}}$ | HyperFace Optimization Runtime |
|---|---|
| 10k | 6 hours |
| 20k | 11 hours |
| 30k | 23 hours |
| 50k | 84 hours |

   We should note that instead of solving the HyperFace optimization on all pairs of points, we can solve the optimization stochastically in which in each iteration a mini-batch of points is considered and optimized. Therefore the complexity will become $\mathcal{O}(b^2)$, where $b$ is size of mini-batch and $b \leq n_{\text{id}}$. This way the complexity of our method can be independent of the number of identities and significantly reduced (especially for $b \ll n_{\text{id}}$). Our stochastic optimization is further studied and discussed in Section B of this Appendix.

2. **Image Generation:** After solving the HyperFace optimization, we need to use the generator network in inference mode and generate the required number of images. Therefore, the generation of dataset has a *linear* complexity with respect to the number of images (i.e., $\mathcal{O}(n_{\text{images}})$, where $n_{\text{images}}$ is the number of images in the generated dataset). The average runtime for generating a single synthetic face image on a system equipped with a single NVIDIA 3090 GPU is 1.25 seconds. For example, generating a dataset with 500,000 images takes about 174 hours on a single NVIDIA 3090 GPU. Note that this optimization process can be parallelized, and therefore image generation can be deployed on a cluster or a farm of GPUs.

## B    HYPERFACE STOCHASTIC OPTIMIZATION

As discussed in Appendix A, HyperFace optimization (Algorithm 1) considers all reference points $\{x_{\text{ref},i}\}_{i=1}^{n_{\text{id}}}$ and has a quadratic complexity $\mathcal{O}(n_{\text{id}}^2)$. To reduce this complexity, in each iteration, we can randomly select a mini-batch of $b$ points and only optimize the selected $b$ reference points instead of all $n_{\text{id}}$ reference points. This way in each iteration we can compare only $\binom{b}{2}$ pairs instead of $\binom{n_{\text{id}}}{2}$ pairs, and therefore the complexity of our optimization will become $\mathcal{O}(b^2)$. In the following, we first theoretically prove that the expected mini-batch gradient approximates the full gradient, and then validate it with experimental analyses.

**Theorem 1.** *Let $X_{ref} = \{x_{ref,i}\}_{i=1}^{n_{id}}$ represent $n_{id}$ points on a $n_{\mathcal{X}}$-dimensional hypersphere $\mathcal{S}$. Consider an objective function:*

$$\mathcal{L}(X_{ref}) = \frac{1}{\binom{n_{id}}{2}} \sum_{i=1}^{n_{id}} \sum_{j=i+1}^{n_{id}} \ell(x_{ref,i}, x_{ref,j}),$$

*where $\ell(\cdot, \cdot)$ denotes a pairwise function. The goal is to minimize $\mathcal{L}(X_{ref})$ for $X_{ref} = \{x_{ref,i}\}_{i=1}^{n_{id}}$. Suppose in each iteration, instead of computing $\nabla\mathcal{L}(X_{ref})$ over all $\binom{n_{id}}{2}$ pairs, we approximate it using a random mini-batch $B \subset X_{ref}$ of size $b \ll n_{id}$. Then, the expected batch gradient approximates the full gradient:*

$$\mathbb{E}[\nabla\mathcal{L}_B(X_{ref})] = \nabla\mathcal{L}(X_{ref}). \tag{4}$$

*Proof.* For a batch $B$ of size $b$, the batch objective is:

$$\mathcal{L}_B(\boldsymbol{X}_{\text{ref}}) = \frac{1}{\binom{b}{2}} \sum_{i \in B} \sum_{j \in B, j > i} \ell(\boldsymbol{x}_{\text{ref},i}, \boldsymbol{x}_{\text{ref},j}). \tag{5}$$

The full gradient of $\mathcal{L}(\boldsymbol{X}_{\text{ref}})$ is:

$$\nabla \mathcal{L}(\boldsymbol{X}_{\text{ref}}) = \frac{1}{\binom{n_{\text{id}}}{2}} \sum_{i=1}^{n_{\text{id}}} \sum_{j=i+1}^{n_{\text{id}}} \nabla \ell(\boldsymbol{x}_{\text{ref},i}, \boldsymbol{x}_{\text{ref},j}). \tag{6}$$

Similarly, the batch gradient is:

$$\nabla \mathcal{L}_B(\boldsymbol{X}_{\text{ref}}) = \frac{1}{\binom{b}{2}} \sum_{i \in B} \sum_{j \in B, j > i} \nabla \ell(\boldsymbol{x}_{\text{ref},i}, \boldsymbol{x}_{\text{ref},j}). \tag{7}$$

The expectation over all possible batches $B$ is:

$$\mathbb{E}[\nabla \mathcal{L}_B(\boldsymbol{X}_{\text{ref}})] = \frac{1}{\binom{b}{2}} \sum_{i=1}^{n_{\text{id}}} \sum_{j=i+1}^{n_{\text{id}}} P[(i,j) \in B] \nabla \ell(\boldsymbol{x}_{\text{ref},i}, \boldsymbol{x}_{\text{ref},j}), \tag{8}$$

where $P[(i,j) \in B]$ is the probability of selecting the pair $(i, j)$ in a random batch. For uniformly sampled random batches:

$$P[(i,j) \in B] = \frac{\binom{b}{2}}{\binom{n_{\text{id}}}{2}} \tag{9}$$

By substituting $P[(i,j) \in B]$ into the expectation in Eq. 8, we will have:

$$\mathbb{E}[\nabla \mathcal{L}_B(\boldsymbol{X}_{\text{ref}})] = \frac{1}{\binom{n_{\text{id}}}{2}} \sum_{i=1}^{n_{\text{id}}} \sum_{j=i+1}^{n_{\text{id}}} \nabla \ell(\boldsymbol{x}_{\text{ref},i}, \boldsymbol{x}_{\text{ref},j}) = \nabla \mathcal{L}(\boldsymbol{X}_{\text{ref}}). \tag{10}$$

Thus, the batch gradient is an unbiased estimator of the full gradient.

$\square$

**Corollary 1.** *A special case for Theorem 1 is when function $\ell(\boldsymbol{x}_{ref,i}, \boldsymbol{x}_{ref,j})$ is defined as follows:*

$$\ell(\boldsymbol{x}_{ref,i}, \boldsymbol{x}_{ref,j}) = \begin{cases} -d(\boldsymbol{x}_{ref,i}, \boldsymbol{x}_{ref,j}) & (i,j) = argmax_{\boldsymbol{X}_{ref}, i \neq j} - d(\boldsymbol{x}_{ref,i}, \boldsymbol{x}_{ref,j}) \\ 0 & otherwise. \end{cases} \tag{11}$$

Therefore, we can rewrite Algorithm 1 with a stochastic optimization as presented in Algorithm 2.

---

**Algorithm 2** HyperFace Stochastic Optimization for Finding Reference Embeddings

---

1: **Inputs**:     $\lambda$ : learning rate, $n_{\text{itr}}$ : number of iterations, $\{\boldsymbol{x}_g\}_{g=1}^{n_{\text{gallery}}}$ : embeddings of a gallery of face images,
2:         $\alpha$ : hyperparameter (contribution of regularization), $b$ : size of mini-batch.
3: **Output**:     $\boldsymbol{X}_{\text{ref}} = \{\boldsymbol{x}_{\text{ref},i}\}_{i=1}^{n_{\text{id}}}$ : optimized reference embeddings.
4: **Procedure:**
5:     Initialize reference embeddings $\boldsymbol{X}_{\text{ref}} = \{\boldsymbol{x}_{\text{ref},i}\}_{i=1}^{n_{\text{id}}}$
6:     **For** $n = 1, .., n_{\text{itr}}$ **do**
7:         Sample a random mini-batch $B \subset \boldsymbol{X}_{\text{ref}}$ of size $b$         ▷ Sampling a random mini-batch
8:         Find $\boldsymbol{x}_{\text{ref},i}, \boldsymbol{x}_{\text{ref},j} \in B$ which have minimum distance $d(\boldsymbol{x}_{\text{ref},i}, \boldsymbol{x}_{\text{ref},j})$
9:         $\text{Reg} \leftarrow \frac{1}{b} \sum_{k=1}^{b} \min_{\{\boldsymbol{x}_g\}_{\text{gallery}}} d(\boldsymbol{x}_{\text{ref},k}, \boldsymbol{x}_g)$         ▷ Calculate the regularization term
10:        $\text{cost} \leftarrow -d(\boldsymbol{x}_{\text{ref},i}, \boldsymbol{x}_{\text{ref},j})$
11:        $B \leftarrow B - \text{Adam}(\nabla \text{cost}, \lambda)$
12:        $B \leftarrow \text{normalize}(B)$     ▷ To ensure that resulting embeddings B remain on the hypersphere.
13:        Update $B$ in $\boldsymbol{X}_{\text{ref}}$
14:     **End For**
15: **End Procedure**

---

To validate our theoretical analyses, we implement the HyperFace stochastic optimization (Algorithm 2) and use the optimized embeddings to generate synthetic face recognition datasets. We

consider 30,000 synthetic identities and solve HyperFace stochastic optimization (Algorithm 2) for different batch sizes. In each case, after solving the stochastic optimization, we generate 50 synthetic images per identity as described in Section 2 (Image Generation). Then, we use the generated datasets to train face recognition models and evaluate the performance of trained face recognition models. Table 10 reports the performance of trained face recognition models. As the results in this table show face recognition models trained with datasets that are generated with stochastic mini-batch optimization achieve comparable performance to the face recognition model trained with the dataset that is generated based on full-batch optimization. Therefore, our experimental results meet our theoretical prediction in Theorem 1.

Table 10: Ablation study on the effect of number of batch size in HyperFace stochastic optimization (Algorithm 2).

| Batch Size | LFW | CPLFW | CALFW | CFP | AgeDB |
|---|---|---|---|---|---|
| 1,000 (mini-batch) | 98.28 | 85.23 | 91.05 | 91.86 | 89.37 |
| 5,000 (mini-batch) | 98.62 | 84.98 | 90.73 | 90.41 | 88.97 |
| 30,000 (full-batch) | 98.38 | 85.07 | 90.88 | 91.57 | 89.60 |

In terms of complexity, the HyperFace stochastic optimization (Algorithm 2) requires significantly less computation resources for solving the optimization. Table 11 reports the runtime for solving the HyperFace stochastic optimization (Algorithm 2) for different batch sizes and different numbers of identities and a fixed size of gallery on a system equipped with a single NVIDIA 3090 GPU. As the results in this table show, the complexity is independent of the number of identities (i.e., $n_{id}$) and depends on the size of mini-batch $b$. Comparing the results in Table 11 and Table 9, we can conclude that our stochastic optimization significantly reduced the complexity.

Table 11: Runtime for solving the HyperFace stochastic optimization (Algorithm 2) for different numbers of identities on a system equipped with a single NVIDIA 3090 GPU.

| Batch Size ($b$) | # ID ($n_{id}$) | HyperFace Stochastic Optimization Runtime |
|---|---|---|
| 1,000 | 30k | 0.4 hours |
| | 50k | 0.5 hours |
| | 100k | 0.5 hours |
| 5,000 | 30k | 2.2 hours |
| | 50k | 2.2 hours |
| | 100k | 2.2 hours |
| 10,000 | 30k | 8.8 hours |
| | 50k | 8.9 hours |
| | 100k | 8.9 hours |

## C  SYNTHETIC DATASETS AT SCALE

In Table 1 of the paper, we compared our face recognition models trained with our generated dataset and synthetic datasets in the literature. For previous datasets, we considered the available version of each dataset which has a similar number of identities (10k). In Table 3, we studied the effect of the number of identities in our dataset generation, where the results showed that we can scale our synthetic dataset and achieve a higher recognition performance. In Table 12, we compare the performance of face recognition models trained with our generated datasets and with all publicly available versions (particularly larger scale) of synthetic datasets in the literature. As the results in this table show, our generated datasets achieve competitive performance with synthetic datasets in the literature at scale. Comparing different datasets in the literature, DCFace, which outperformed previous datasets in Table 1, does not achieve the best performance on any of the benchmarks for its larger version. In contrast, Langevin-DisCo achieves a significant improvement for its larger version with 30k identities compared to its smaller version with 10k identities. However, Geissbühler et al. (2024) reported a lower performance for their dataset with 50k identities compared to 30k identities, indicating limitations in further scaling the Langevin-DisCo dataset for more than 30k identities.

Table 12: Comparison of recognition performance of face recognition models trained with *the largest available versions* of different synthetic datasets as well as a real dataset (i.e., CASIA-WebFace). The performance reported for each dataset is in terms of accuracy and best value for each benchmark is emboldened.

| Dataset name | # IDs | # Images | LFW | CPLFW | CALFW | CFP | AgeDB |
|---|---|---|---|---|---|---|---|
| SynFace (Qiu et al., 2021) | 10'000 | 999'994 | 86.57 | 65.10 | 70.08 | 66.79 | 59.13 |
| SFace (Boutros et al., 2022) | 10'572 | 1'885'877 | 93.65 | 74.90 | 80.97 | 75.36 | 70.32 |
| Syn-Multi-PIE (Colbois et al., 2021) | 10'000 | 180'000 | 78.72 | 60.22 | 61.83 | 60.84 | 54.05 |
| IDnet (Kolf et al., 2023) | 10'577 | 1'057'200 | 84.48 | 68.12 | 71.42 | 68.93 | 62.63 |
| ExFaceGAN (Boutros et al., 2023b) | 10'000 | 599'944 | 85.98 | 66.97 | 70.00 | 66.96 | 57.37 |
| GANDiffFace (Melzi et al., 2023) | 10'080 | 543'893 | 94.35 | 76.15 | 79.90 | 78.99 | 69.82 |
| Langevin-Dispersion (Geissbühler et al., 2024) | 10'000 | 650'000 | 94.38 | 65.75 | 86.03 | 65.51 | 77.30 |
| Langevin-DisCo (Geissbühler et al., 2024) | 10'000 | 650'000 | 97.07 | 76.73 | 89.05 | 79.56 | 83.38 |
| Langevin-DisCo (Geissbühler et al., 2024) | 30'000 | 1'950'000 | **98.97** | 81.52 | **93.95** | 83.77 | **93.32** |
| DigiFace-1M (Bae et al., 2023) | 109'999 | 1'219'995 | 90.68 | 72.55 | 73.75 | 79.43 | 68.43 |
| IDiff-Face (Uniform) (Boutros et al., 2023a) | 10'049 | 502'450 | 98.18 | 80.87 | 90.82 | 82.96 | 85.50 |
| IDiff-Face (Two-Stage) (Boutros et al., 2023a) | 10'050 | 502'500 | 98.00 | 77.77 | 88.55 | 82.57 | 82.35 |
| DCFace (Kim et al., 2023) | 10'000 | 500'000 | 98.35 | 83.12 | 91.70 | 88.43 | 89.50 |
| DCFace (Kim et al., 2023) | 60'000 | 1'200'000 | 98.90 | 84.97 | 92.80 | 89.04 | 91.52 |
| **HyperFace [ours]** | 10'000 | 640'000 | 98.67 | 84.68 | 89.82 | 89.14 | 87.07 |
| **HyperFace [ours]** | 50'000 | 3'200'000 | 98.27 | **85.60** | 91.48 | **92.24** | 90.40 |
| CASIA-WebFace (Yi et al., 2014) | 10'572 | 490'623 | 99.42 | 90.02 | 93.43 | 94.97 | 94.32 |

Nevertheless, our method achieves improvement by scaling the number of identities (Table 3). In particular, our dataset with 50k identities and 3.2M images achieves competitive performance with large-scale synthetic datasets in the literature.

## D    IDENTITY LEAKAGE AND RECOGNITION PERFORMANCE

In Section 3.3, we discussed identity leakage in the generated face datasets. While the leakage of identity is not evident in the generated dataset, it is important to see if identity leakage may affect the recognition performance of face recognition models. To this end, we consider the FFHQ and CASIA-WebFace datasets as two real face datasets and compare all possible pairs from our synthetic dataset with images in the real datasets. Then, for each of these real datasets, we find the top-200 pairs (synthetic-real) and exclude the corresponding synthetic image from our generated dataset. This ensures that images which may contain identity leakage are excluded from the final synthetic datasets. We use the resulting cleaned datasets to train face recognition models and compare them with the face recognition model trained on our original synthetic dataset. Table 13 reports the recognition performance of face recognition models trained with original and cleaned synthetic datasets.

Table 13: Evaluation of potential identity leakage on the recognition performance.

| Synthetic Dataset | LFW | CPLFW | CALFW | CFP | AgeDB |
|---|---|---|---|---|---|
| cleaned (Ref.: CASIA-WebFace) | 98.52 | 84.80 | 89.52 | 89.43 | 87.00 |
| cleaned (Ref.: FFHQ) | 98.77 | 84.53 | 89.35 | 89.36 | 86.67 |
| original | 98.67 | 84.68 | 89.82 | 89.14 | 87.07 |

As the results in Table 13, removing images with similar identities does not impact the recognition performance of the trained face recognition model. However, we would like to highlight that while identity leakage may not affect recognition performance on benchmark datasets, it is an important privacy concern.

## E    ADDITIONAL ABLATION STUDY

In Section 3, we reported ablation studies on different hyperparameters in our dataset generation. As a new experiment, we consider different optimizers for solving HyperFace optimization (full batch). We consider RMSprop, Adam, and AdamW optimizers. Table 14 compares the performance of the face recognition model trained with datasets that are generated based on different optimizers in HyperFace optimization. As the results in this table show, solving HyperFace optimization with different optimizers leads to comparable performance.

Table 14: Ablation study on optimizer

| Optimizer | LFW | CPLFW | CALFW | CFP | AgeDB |
|-----------|-------|-------|-------|-------|-------|
| RMSprop | 98.47 | 84.73 | 89.18 | 89.27 | 86.83 |
| AdamW | 98.70 | 84.20 | 89.02 | 89.41 | 86.38 |
| Adam | 98.67 | 84.68 | 89.82 | 89.14 | 87.07 |

As another experiment, we generate random points on the hypersphere and use random points as reference embeddings without HyperFace optimization to generate a synthetic dataset. We ensure that selected points have at least 0.3 cosine distance. Table 15 compares the performance of the face recognition model trained with the dataset based on random embeddings and HyperFace optimization. As the results in this table show, solving HyperFace optimization achieves superior performance on all benchmarks. Note that with a random selection of points on the hypersphere, there is no guarantee to be on the manifold of embeddings of the face recognition model. However, with our HyperFace optimization, we try to keep points on the face recognition manifold, which results in a dataset that leads to better performance.

Table 15: Ablation study on using random points vs HyperFace optimization.

| Reference Embeddings | LFW | CPLFW | CALFW | CFP | AgeDB |
|----------------------|-------|-------|-------|-------|-------|
| Random | 98.12 | 83.67 | 86.67 | 88.79 | 84.25 |
| HyperFace Optimization | **98.67** | **84.68** | **89.82** | **89.14** | **87.07** |

## F    HYPERFACE DATASET GENERATION

We described HyperFace dataset generation in 2. Algorithm 3 summarizes the dataset generation process in our method.

---
**Algorithm 3** HyperFace Dataset Generation

---
1: **Inputs**:    $n_{\text{id}}$ : number of synthetic identities, $n_{\text{sample}}$ : number of sample images per identity,
2:          $G$ : face generator model, $\beta$ : hyperparameter (controls variations in embeddings)
3: **Output**:    $\mathcal{D}_{\text{HyperFace}} = \{I\}$ : generated dataset.
4: **Procedure:**
5:    Solve HyperFace optimization to find reference embeddings $X_{\text{ref}} = \{x_{\text{ref},i}\}_{i=1}^{n_{\text{id}}}$      ▷ Algorithm 1 or 2
6:    Initialize $\mathcal{D}_{\text{HyperFace}}$= [ ]
7:    **For** $x_{\text{ref}} \in \{x_{\text{ref},i}\}_{i=1}^{n_{\text{id}}}$ **do**
8:      **For** $n = 1, .., n_{\text{sample}}$ **do**
9:        Sample Gaussian noise $z \sim \mathcal{N}(0, \mathbb{I}^{\text{DM}})$                          ▷ For diffusion model $G$
10:        Sample Gaussian noise $v \sim \mathcal{N}(0, \mathbb{I}^{n_\mathcal{X}})$                ▷ For variations in the embedding $x_{\text{ref}}$
11:        Generate synthetic image $I = G(\frac{x_{\text{ref}}+\beta v}{||x_{\text{ref}}+\beta v||_2}, z)$
12:        $\mathcal{D}_{\text{HyperFace}}$.append($I$)                  ▷ Store the generated image $I$ in the dataset $\mathcal{D}_{\text{HyperFace}}$
13:      **End For**
14:    **End For**
15: **End Procedure**

---

# G VISUALIZATION

Figure 4 illustrates sample face images from the HyperFace dataset. In addition, Figure 5 and Figure 6 also show intra-class variations for two synthetic identities in the HyperFace dataset.

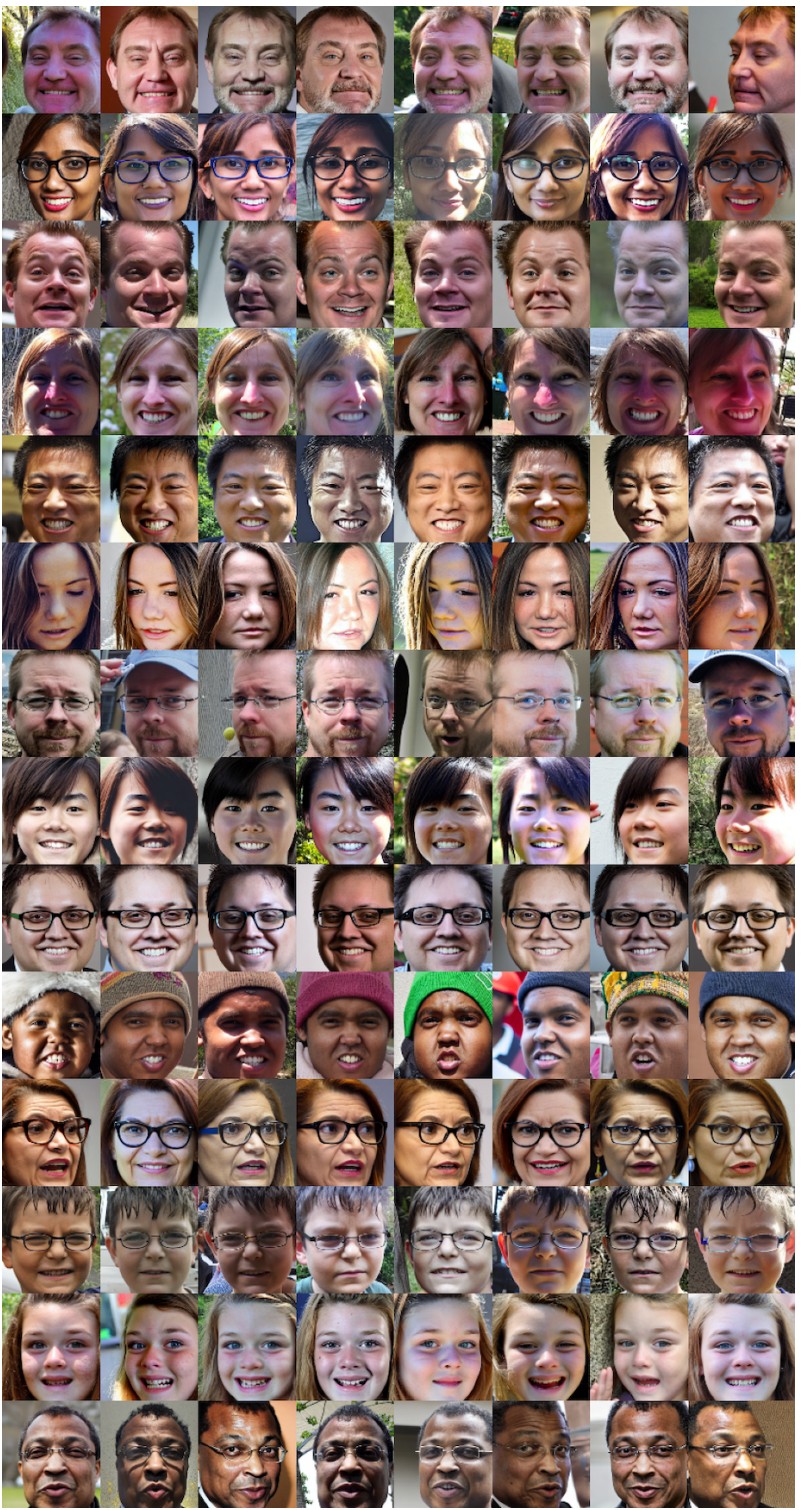

Figure 4: Sample face images of different synthetic identities from the HyperFace dataset.

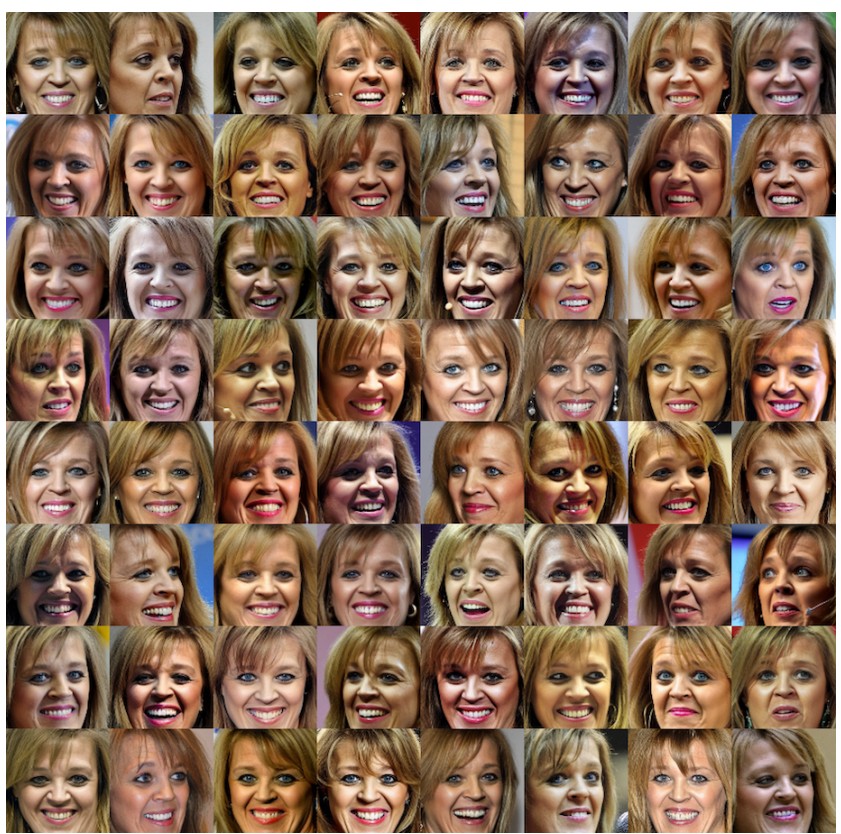

Figure 5: Sample face images of one subject from the HyperFace dataset (intra-class variations).

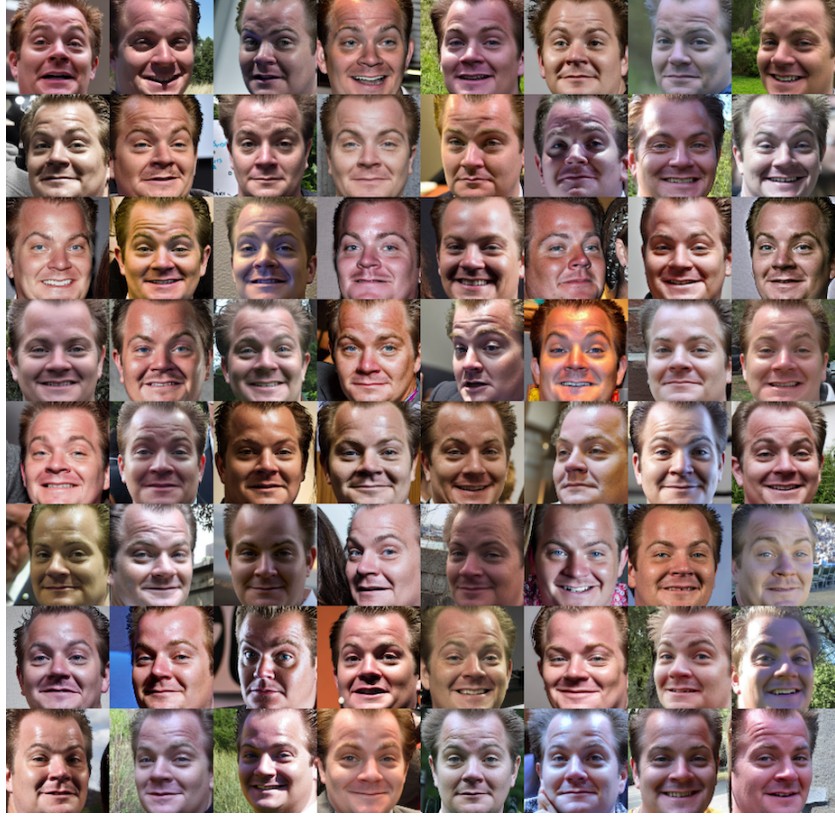

Figure 6: Sample face images of one subject from the HyperFace dataset (intra-class variations).

