# OpenReview forum: "HyperFace: Generating Synthetic Face Recognition Datasets by Exploring Face Embedding Hypersphere"
_ICLR.cc/2025/Conference — ICLR 2025 Poster_

### Official Review · Reviewer_RHfp · 2024-10-27

**Soundness:** 4
**Presentation:** 3
**Contribution:** 3
**Rating:** 5
**Confidence:** 5

**Summary:**

This paper proposes a synthetic data generation method for FR that aims to improve inter-class variation compared to existing methods. The approach utilizes the embedding space of a pretrained FR model to create an initial gallery, then optimizes these embeddings to uniformly position identities on a unit hypersphere. A conditional diffusion generator is subsequently used to synthesize faces.

**Strengths:**

1. The paper is well-written and easy to follow.
2. The empirical study shows that the proposed method generalizes well across different validation sets.
3. Experimental results are promising, with satisfactory improvements on benchmark datasets.

**Weaknesses:**

1. HyperFace Optimization: Figure 2 shows that HyperFace optimization results in a uniform distribution of points on the hypersphere. However, the paper lacks evidence, analysis, or experiments demonstrating that Equation 1 is minimized by this uniform distribution. See [1] for more details.

2. Use of Hyperspherical Points: One key aspect of the method is the use of uniformly positioned points on the hypersphere. While using a pre-trained FR model for generating the identity gallery is reasonable, it would be helpful to see results without applying HyperFace (using pure FR embedding as condition). How would this affect the results?

3. Experimental Section: The experiments section needs major revisions. Most of the subsections only report table results without in-depth discussion or analysis. For a conference of ICLR’s quality, it’s important to explain specific behaviors and insights from the proposed method.

4. Additional Experiments: Additional experiments could improve clarity on the benefits of the method. For example, while Table 3 presents an ablation study on the number of identities on FR performance, it would also be valuable to show how many novel identities the method can generate (what is the saturation point for the number of novel identities).

5. Identity Leakage: The paper mentions identity leakage but lacks in-depth experiments on the synthesized data. What would the performance look like if synthesized images with high similarity to real datasets (e.g., CASIA) were excluded?

**Questions:**

My main concerns are threefold: (1) the lack of theoretical and empirical analysis on HyperFace optimization (Equation 1), (2) missing ablations and detailed analysis of results, and (3) understanding the limitations of the method in generating novel identities. I am hopeful these concerns can be addressed during the discussion period, and I am open to increasing my score based on the responses. Please also see the weaknesses

---

> ### Author Response · Authors · 2024-11-21
> **Authors Reply to Reviewer RHfp [Part 1/2]**
>
> We thank the reviewer for their time in reviewing our paper and for their valuable comments. We are happy that the reviewer found our paper *well-written and easy to follow*. We are also delighted for the reviewer's positive feedback on the *generalization across different validation sets* of our method with *promising satisfactory improvements on benchmark datasets*.
> Below, we tried our best to address the concerns raised by the reviewer:
>
> - **Reply to Weakness #1 (Uniform Distribution)**: While equation (1) is focused on optimization over the entire hypersphere (n-ball), the face recognition manifold does not necessarily cover the entire hypersphere. For this reason, we proposed a regularization in equation (2) which tries to keep our HyperFace optimization on the manifold of the face recognition using embeddings from embeddings from a gallery of face images. Therefore, we cannot theoretically study the distribution of points on the entire hypersphere, and instead we need to consider the distribution of points on the manifold of the face recognition over hypersphere, which is not trivial and requires accurate estimation of the manifold.
>
>
> - **Reply to Weakness #2 (HyperFace Optmimization)**:  Following the reviewer's suggestion, we added an ablation study in Appendix E and studied the case where random points are used as reference embeddings for generating dataset (i.e., without HyperFace optimization). As the results in our new experiment in Appendix E show, our HyperFace optimization results in a synthetic dataset with better quality which achieves superior performance.
>
>
> - **Reply to Weakness #3 (Experiments and Discussions)**: We tried to include an extensive study on different parameters in our paper. Following the reviewer's comment we revised our experiments and further discussed our results. In the revised version of the paper, we carefully considered all experiments suggested by reviewers and expanded our analyses. We also explored the complexity of our dataset generation in Appendix A. We proposed a stochastic optimization, which reduces the complexity, and provided in-depth theoretical and experimental analyses in Appendix B. We also provided more experiments in the appendix of the revised version of the paper (Appendix C: synthetic datasets at scale, Appendix D: idenitity leakage and recognition performance, Appendix E: additional ablation study). In case any part remains unclear, we appreciate it if the reviewer can clarify the missing part and the required discussion.
>
>
> - **Reply to Weakness #4 (Additional Experiment)**: Following the reviewer's suggestion, we added another experiment, in which we increased the number of identities to 50,000 in Table 3. The results indicate improvements in the performance of our model over different benchmarks. Therefore, we cannot conclude saturation on 50,000 identities, and further experiments are required.
>
> | # IDs | LFW | CPLFW | CALFW | CFP | AgeDB |
> |---|:---:|:---:|:---:|:---:|:---:|
> | 10k | 98.67 | 84.68 | 89.82 | 89.14 | 87.07 |
> | 30k | 98.82 | 85.23 | 91.12 | 91.74 | 89.42 |
> | 50k | 98.27 | 85.6 | 91.48 | 92.24 | 90.4 |
>
> While increasing the number of identities requires more computation time, we should note that our new generated dataset with 50,000 identities has already the largest number of synthetic images compared to all synthetic datasets in the literature in Table 1. We should note that we also added a new section in appendix (Appendix C), where we compared our method with largest synthetic datasets in the literature. Our dataset with 50k identity and 3.2M is larger than synthetic datasets compared from the literature and achieves competitive performance with previous large-scale synthetic datasets as reported in Appendix C.
>
>
> - **Reply to Weakness #5 (Identity Leakage)**: We thank the reviewer for raising this point and suggesting this interesting experiment. Following the reviewer's suggestion, we conducted a new experiment in which we excluded images which have high similarity with real images. Then, we use the cleaned dataset to train a new face recognition model and benchmark the performance of the trained face recognition model. The results show that the new face recognition models achieve comparable recognition performance. We added a new section to Appendix D and discussed it in detail.

---

> > ### Comment · Reviewer_RHfp · 2024-11-22
> > **I maintain my rating.**
> >
> > Thanks to the author for providing feedback on my comment. Some points in my initial review have yet to be addressed.
> > 1- In lines 137-148 and Equation 1 of the original manuscript, the paper tries to maximize the minimum distance of the randomly selected pairs. Intuitively, pushing all features away from each other should indeed cause them to be roughly uniformly distributed. Please refer to the [2] for more details.
> > 2- Please note that based on the authors' (50K identities), the proposed method is not the largest synthetic FR dataset. DigiFace [1], containing 1.22M images of 110K identities, is the largest public synthetic dataset for face recognition.
> > 3- What is the maximum number of unique identities that the proposed method can produce?  Please see Figure 3 in [3] for clarification.
> >
> > [1]Bae, Gwangbin, et al. "Digiface-1m: 1 million digital face images for face recognition." Proceedings of the IEEE/CVF Winter Conference on Applications of Computer Vision. 2023.
> > [2]Wang, Tongzhou, and Phillip Isola. "Understanding contrastive representation learning through alignment and uniformity on the hypersphere." International conference on machine learning. PMLR, 2020.
> > [3]Kim, Minchul, et al. "Dcface: Synthetic face generation with dual condition diffusion model." Proceedings of the ieee/cvf conference on computer vision and pattern recognition. 2023.

---

> ### Author Response · Authors · 2024-11-21
> **Authors Reply to Reviewer RHfp [Part 2/2]**
>
> We think that three questions raised by the reviewer are potentially answered in our reply to the raised weaknesses:
> - **Reply to Question #1**: Please see our reply to weakness #1 about uniform distribution and face recognition manifold which makes theoretical analysis difficult. We would like to note that in the revised version of the paper, we further explored the complexity of our optimization in Appendix A and proposed a stochastic optimization that reduces the complexity in Appendix B. We provide thorough theoretical and experimental analyses where our experimental results meet our theoretical predictions.
> - **Reply to Question #2**: Please see our reply to weakness #2, #3, and #4. We added a larger dataset with 50k identities in Table 3 and also provided more experiments in the appendix of the revised version of the paper (Appendix C: synthetic datasets at scale, Appendix D: idenitity leakage and recognition performance, Appendix E: additional ablation study).
> - **Reply to Question #3**: Please see our reply to weakness #5.
>
> We hope we could adequately address the reviewer's concerns.
> In case the reviewer found our reply convincing, we appreciate it if the reviewer can increase their scores and rating.
> We are happy to continue the discussion if any part remains unclear.

---

> ### Author Response · Authors · 2024-11-24
> **Authors Reply to Reviewer feedback**
>
> We appreciate the reviewer for their engagement in the discussion. We are happy that our reply could address some of the initial concerns. Below, we tried to address the new points mentioned by the reviewer:
>
> - **Reply to Point #1 (Uniform Distribution):** As mentioned earlier in our **Reply to Weakness #1 (Uniform Distribution)**, our method does not only include optimizing equation (1), but we have an additional regularization term in our optimization in equation (2). **The regularization term in equation (2) prevents uniform distribution of points over the hypersphere**. In fact, the regularization term tries to keep the solution space for our optimization close to the manifold of face recognition embeddibgs on the hypersphere instead of the entire surface of the hypersphere. Therefore, **equation (2) cannot result in a uniform distribution over the entire hypersphere**. Our ablation study in Table 6 also shows that our regularization in equation (2) improves the recognition performance of the trained face recognition model.
>
> - **Reply to Point #2 (largest dataset):** Our dataset with 50k identities has 3.2M images, which is the largest dataset in terms of *number of images*. As mentioned by the reviewer, DigiFace has 110k identities but with 1.22M images. While we acknowledge that DigiFace has more *identities*, in our **Reply to Weakness #4 (Additional Experiment)** we stated that *our new generated dataset with 50,000 identities has already the largest `number of synthetic images`* (i.e., 3.2M). We should note that our datasets (with 50k identities and even 10k identities) far outperform DigiFace on all benchmarks. In Appendix C, we also provided a benchmark with the largest available version of all datasets, where our dataset achieves competitive performance with state-of-the-art in the literature. Please note that in the paper, we have not mentioned that our dataset is the largest dataset in the literature.
>
> - **Reply to Point #3 (Maximum Number of Unique Identities):**
> We would like to note that Figure 3 of reference [3] (i.e., DCFace paper) mentioned by the reviewer is on the capacity of face generator models (DiscoFaceGAN and unconditional DDPM). In fact **this figure does not provide an answer to the maximum number of unique identities that can be generated by DCFace**. In particular, for Figure 3 in the DCFace paper, Kim *et al.* [3] generated 10,000 images with two face generator models (DiscoFaceGAN and DDPM) and compared all the generated images. By changing the threshold of the face recognition model, they plotted Figure 3 in [3]. Therefore, even in the DCFace paper (CVPR 2023) the question of the *maximum number of unique identities* has not been addressed. While in Figure 3, a maximum of 10k identities are counted, Kim et al. [3] also published a version of the dataset with 60k identities. Therefore, the maximum number of identities that can be generated by DCFace is not evident in Figure 3 of the DCFace paper [3].
> As a matter of fact, the *maximum number of unique identities* is more studeied for the capacity of face generator models, such as [A].
> However, we believe for answering to this question for synthetic datasets, we need to increase the number of identities in generated datasets and train face recognition with larger datasets. As reported in the paper, by increasing the number of identities, we still could not observe saturation in performance, and therefore we cannot ensure the maximum number of unique identities in our method. However, generating a larger dataset requires more computation and time, while our method with 50k identity already achieves competitive performance with state of the art.
>
> [A] Boddeti, et al. "On the biometric capacity of generative face models", IEEE International Joint Conference on Biometrics (IJCB), 2023.
>
> We would be glad to continue the discussion if any part remains unclear.

---

> > ### Comment · Reviewer_RHfp · 2024-11-25
> > **please see my comment**
> >
> > Thanks to the authors for addressing my concerns. I understand that the regularization term of equation 2 prevents the uniform distribution. My point concerns equation 1 and its optimization( lines 105-148) which lacks theoretical and empirical analysis. papers solely says "we solve the optimization problem with an iterative approach based on gradient descent" and does not provide insight into the optimization and the cost function for it.

---

> > > ### Author Response · Authors · 2024-11-26
> > > **Authors Reply to Reviewer Feedback**
> > >
> > > We thank the reviewer for their reply and their continued engagement in the discussion. We are happy that our reply could address the reviewer's concerns. Below, we tried to address the remaining concern of the reviewer:
> > >
> > > Our loss function in equation (1) represents a famous *open* problem, which is known as spherical code optimization or Tammes problem.
> > > The optimal solutions for the Tammes problem are studied for small dimensions and a small number of points. However, for large dimensions and a high number of points there is no closed-form solution for the Tammes problem. There are different approaches for solving this optimization problem (such as geometric optimization, numerical optimization, etc.) for large dimensions and a high number of points. However, for a large dimension (e.g., 512) and a *very* large number of points (e.g., 10k identities) solving this problem with geometric optimization or numerical optimization is computationally very expensive. Hence, we solve this problem with a gradient descent approach which allows us to solve the optimization with a reasonable computation resource. In Appendix A, we report the computation required to solve our optimization with our method and further reduce the computation with a stochastic optimization in Appendix B, where we demonstrated theoretically and empirically that stochastic optimization reduces the complexity while resulting in a comparable performance.
> > >
> > > Because the Tammes problem is very difficult for large dimensions and a high number of points, an in-depth analysis of equation (1) requires extensive study. We would like to refer the reviewer to a very recent ICML 2024 workshop paper [B], which is focused only on the Tammes problem, and supports the reviewer's intuition of (near) uniform distribution of the (sub) optimum solutions. Given the sophistication and difficulty of the Tammes problem for high dimensions and the fact that our final loss function is indeed equation (2), we believe further analysis of equation (1) is out of the focus of this paper. However, to address the reviewer's concern we updated the paper and elaborated further on the previous studies on the Tammes problem in the second revised version of the paper.
> > >
> > > [B] Tokarchuk, et al. "On the Matter of Embeddings Dispersion on Hyperspheres", In ICML 2024 Workshop on Geometry-grounded Representation Learning and Generative Modeling, 2024. URL: https://openreview.net/pdf?id=yh4IjSjAhQ
> > >
> > > We sincerely appreciate the reviewer's comments, both in the initial review and discussions, which helped us improve the quality of our paper. We are, of course, gladly open to further discussions if the reviewer still has any concerns.

---

### Official Review · Reviewer_YfZL · 2024-10-30

**Soundness:** 3
**Presentation:** 2
**Contribution:** 1
**Rating:** 5
**Confidence:** 5

**Summary:**

This paper propose an interesting solution for synthetic face recognition, i.e. optimizing the hyperspace for generation. The solution is to treat the face generation as a packing problem on the embedding space. The optimization goal is to find feature embedding that is of high inter-class variation. Finally this paper adopt a pretrained generation method to generate the dataset.

**Strengths:**

1. This paper is good in writing and has a solid mathematical formulation.

**Weaknesses:**

1. [Major] I don't see the motivation to convert the face generation problem as a packing problem on the embedding space from the storytelling, Please provide some related empirical/ theoretical works regarding why the packing problem could be of use for SFR.

2. [Major] This paper adopts Arc2Face for final image generation. However, (1) The ablation study doesn't show the advance of directly sending random feature embedding (each embedding is ensured by restricting the similarity below a certain threshold, e.g. 0.3) to Arc2Face; (2) The comparison with Arc2Face is missing in Table 1, additionally the experiment is marginal better than DCFace. The average performance is 89.876 which is similar to DCFace and dramatically lower than Arc2Face.

3. [Major] In Section 'Solving the HyperFace Optimization', the authors choose AdamW for the optimization solution. However, the other alternative optimization methods are not specified and compared in this paper.

4. Another concern is that the proposed method generates more images(640k) to produce similar performance to DCFace (500k).

5. large intra-class variations can not be observed in the visualization section.

6. [Minor] Notation is not specified in fig 1.2. Please provide more description for the reader to understand the mathematical formulation and the whole generation process. For example, what does reference embedding stand for, I understood it only when I saw the 'Image Generation' section. And what is X_{g}?

7. Please give some detailed pseudo-code for the entire process(training/ generation) for the reader to understand the method.

**Questions:**

Please see the weakness.

If the authors address the concerns well, I am happy to raise my score.

---

> ### Author Response · Authors · 2024-11-21
> **Authors Reply to Reviewer YfZL [Part 1/2]**
>
> We thank the reviewer for their time in reviewing our paper and for their valuable comments. We are happy that the reviewer found our paper *"good in writing and has a solid mathematical formulation"*. Below, we tried our best to address the concerns raised by the reviewer:
>
> - **Reply to Weakness #1 [Major] (motivation)**: In general, in each face recognition dataset (either synthetic or real), the variations in images are very important: especially, it is necessary to have a diversity in the identity (inter-class variation) and also variations for images of each subject (intra-class variation). As a matter of fact, having sufficient variations in a dataset has a direct impact on the generalization capability of the face recognition model (trained with that dataset) and its performance on different benchmarks. So, it can be useful to see how we can represent a face recognition dataset in a loower dimension (compared to image dimensions) and then see how we can use such representation to improve synthetic dataset generation.
> Previous work, e.g., [1,2], showed that we can study face images on embedding space of a face recognition model. The normalized embedding space of a pretrained face recognition model can shape a hypersphere (i.e., n-ball), and we can represent the face dataset on the hypersphere by using the embeddings of images. In order to have a larger variation in a dataset, we would like the face embeddings to cover the most areas on the hypersphere. Therefore, our question is how to distribute identities (using their embeddings) on the face hypersphere. This leads to a packing problem to find an optimum representation for a given number of identities. While representing face images on the hypersphere of a pretrained face recognition model has been used and studied in the literature [1,2], to our knowledge our paper is the first work that formulates the identity sampling for synthetic dataset generation as a packing problem.
> We would like to stress that intra-class variation can be improved by using conditional face generator models (for pose, light conditions, etc.) when generating the synthetic dataset. However, still the major problem is how to increase inter-class variation, which is the focus of our paper and we tried to see how we can improve inter-class variation assuming we can sample identites from face embedding hypersphere.
>   - [1] Terhörst, et al. "On the (limited) generalization of masterface attacks and its relation to the capacity of face representations", IEEE International Joint Conference on Biometrics (IJCB), 2022.
>   - [2] Boddeti, et al. "On the biometric capacity of generative face models", IEEE International Joint Conference on Biometrics (IJCB), 2023.
>
>
>
> - **Reply to Weakness #2 [Major] - Point 2 (Comparison with Arc2Face - Table 1)**: For a fair comparison in our experiments for Table 1, we fixed all hyperparameters and trained face recognition models for all datasets with the same configuration (backbone, loss function, batch size, number of epochs, etc.). Therefore, we could only compare with methods that have publicly available datasets. Whereas the authors of Arc2Face have not published their dataset, and we could not reproduce the results reported in their paper. There are also open issues on their GitHub repository, where several researchers reported that they could not reproduce their results for face recognition. Since our training setup is different, we believe the numbers reported in our experiments are not comparable with the values reported in the Arc2Face paper. The configuration for training face recognition in Arc2Face paper is neither available, which makes it more difficult to make a fair comparison.
>
> - **Reply to Weakness #2 [Major] - Point 1 (Comparison with Arc2Face - Ablation study)**: Following the reviewer's suggestion, we conducted a new experiment and used random embeddings (without HyperFace optimization) to generate a synthetic dataset. We also ensure that the similarity of each pair is below a certain threshold (0.3). As the results in Appendix E show,  our HyperFace optimization results in a synthetic dataset with better quality which achieves superior performance.

---

> ### Author Response · Authors · 2024-11-21
> **Authors Reply to Reviewer YfZL [Part 2/2]**
>
> - **Reply to Weakness #3 [Major] (Optimization)**: Following the reviewer's suggestion, we conducted a new experiment and used other optimization techniques. We added the results to in Appendix E of the revised version of the paper. As the results in our new ablation study show, solving HyperFace optimization with different optimizers leads to comparable performance.
>
> - **Reply to Weakness #4 (Number of Images)**: While by default, we generated 64 samples per identity, we reported an ablation study for different numbers of samples per identity, including 50 samples. Comparing the results reported in the ablation study with Table 1, we can see that still with 50 samples per identity, our method achieves comparable performance. Considering the reviewer's concern, we updated the results in Table 1 with 50 samples per identity (500k images). However, still with 500k images, the ranking of synthetic datasets over different benchmarks remains unchanged.
>
> - **Reply to Weakness #5  (Visualization for Intra-class Variations)**: In Figure 1 of the paper and also Figure 4 of appendix, we illustrated sample face images from our dataset, including 8 samples for each subject to illustrate the intra-class variation. Following the reviewer's comment, in the revised version of the paper, we added 64 images of two synthetic identities in Appendix G to illustrate intra-class variations in our dataset.
>
> - **Reply to Weakness #6 [Minor] (Caption of Fig 2)**: Figure 2 provides a general overview of our method. Following the reviewer's suggestion, we extended the caption and provided more details of our data generation in the caption of this figure.
>
> - **Reply to Weakness #7 (Pseudo-code for the entire process)**: Following the reviewer's suggestion, we added a pseudo-code for the entire data generation process in Appendix F of the revised version of the paper.
>
>
> We hope we could adequately address the reviewer's concerns.
> In case the reviewer found our reply convincing, we appreciate it if the reviewer can increase their scores and rating.
> We are happy to continue the discussion if any part remains unclear.

---

> > ### Comment · Reviewer_YfZL · 2024-11-26
> >
> > Thank the authors for their efforts in addressing my concerns. However, I still have the following issues, which prevent me from improving my rating:
> >
> > (1) 500K Data Setting:
> > The average performance in the 500K data setting is 89.498, which is lower than DCFace's performance listed in the table (90.22). This discrepancy raises concerns about the effectiveness of the proposed approach under this setting.
> >
> > (2) Visualization:
> > The proposed method still lacks sufficient age-related variation in its visualizations, which is an important aspect that remains unaddressed.

---

> > > ### Author Response · Authors · 2024-11-26
> > > **Authors Reply to Reviewer Feedback [Part 1/2]**
> > >
> > > We appreciate the reviewer for their engagement in the discussion. We are happy that our reply could address some of the initial concerns, including major concerns. Below, we tried to address the points mentioned by the reviewer in their new feedback:
> > >
> > > **Reply to Point #1 (Comparison):** Comparing our dataset with 500,000 images against DCFace in Table 1, we can observe that the face recognition model trained with our dataset achieves superior performance on 3 [out of 5] benchmarks. However, our method has inferior performance on AgeDB and CALFW (Cross-Age LFW), which leads to an overall drop in the average accuracy, as raised by the reviewer. However, both AgeDB and CALFW are focused on the evaluation of face recognition models for age variations. We further discuss the limitation of our method for age variations in our **Reply to Point #2 (Visualization: Age-related Variations). However, we would like to argue that comparing the average accuracies over all benchmarks may not lead to a fair comparison. As can be seen in benchmarks (e.g., in Table 1) the variation ranges of accuracies over different benchmark datasets is not constant. For example, differences in LFW is very competitive and it is difficult to improve the performance of LFW by 1%, while the variations are much larger for AgeDB. Therefore, if we consider average accuracy, we are inherently putting more weight on datasets with higher variations. For this reason, in several challenges on face recognition models, such as EFaR@IJCB2023, SDFR@FG2024, etc., the Borda Count has been for leaderboards, instead of average accuracy. The Borda Count considers the rank of each model on each benchmark separately, and then averages the points achieved by rankings over all datasets.
> > >
> > > We would also like to highlight that even if we consider the average accuracy on all datasets, our method achieves the *best average accuracy* when compared with large synthetic datasets in the literature. In Appendix C, we compared face recognition models with larger versions of each dataset (which are publicly available), where our dataset with 50k identities and 3.2M images achieves competitive performance with literature. Meanwhile, if we calculate the average accuracies over all benchmarks (as reported in the following table) our dataset with 50k identities has the *highest average accuracy* compared to large synthetic datasets in the literature:
> > >
> > >
> > > | Dataset Name                                   | # IDs   | # Images  | Average Accuracy (%)|
> > > |------------------------------------------------|---------|-----------|---------|
> > > | SynFace [Qiu et al., 2021]                     | 10,000  | 999,994   | 69.53   |
> > > | SFace [Boutros et al., 2022]                   | 10,572  | 1,885,877 | 79.04   |
> > > | Syn-Multi-PIE [Colbois et al., 2021]           | 10,000  | 180,000   | 63.13   |
> > > | IDnet [Kolf et al., 2023]                      | 10,577  | 1,057,200 | 71.12   |
> > > | ExFaceGAN [Boutros et al., 2023]               | 10,000  | 599,944   | 69.46   |
> > > | GANDiffFace [Melzi et al., 2023]               | 10,080  | 543,893   | 79.84   |
> > > | Langevin-Dispersion [Geissbühler et al., 2024] | 10,000  | 650,000   | 77.79   |
> > > | Langevin-DisCo [Geissbühler et al., 2024]      | 10,000  | 650,000   | 85.16   |
> > > | Langevin-DisCo [Geissbühler et al., 2024]      | 30,000  | 1,950,000 | 90.31   |
> > > | DigiFace-1M [Bae et al., 2023]                 | 109,999 | 1,219,995 | 76.97   |
> > > | IDiff-Face (Uniform) [Boutros et al., 2023]    | 10,049  | 502,450   | 87.67   |
> > > | IDiff-Face (Two-Stage) [Boutros et al., 2023]  | 10,050  | 502,500   | 85.85   |
> > > | DCFace [Kim et al., 2023]                      | 10,000  | 500,000   | 90.22   |
> > > | DCFace [Kim et al., 2023]                      | 60,000  | 1,200,000 | 91.45   |
> > > | **HyperFace [Ours]** | 10,000  | 640,000   | 89.88   |
> > > | **HyperFace [Ours]** | 50,000  | 3,200,000 | **91.60** |
> > >
> > > References:
> > > - EFaR@IJCB2023 competition summary paper: "EFaR 2023: Efficient face recognition competition", In Proc. of the IEEE International Joint Conference on Biometrics (IJCB), 2023.
> > > - SDFR@FG2024 competition summary paper: "SDFR: Synthetic data for face recognition competition", In Proc. of the IEEE 18th International Conference on Automatic Face and Gesture Recognition (FG), 2024.

---

> > > ### Author Response · Authors · 2024-11-26
> > > **Authors Reply to Reviewer Feedback [Part 2/2]**
> > >
> > > **Reply to Point #2 (Visualization: Age-related Variations):** We acknowledge the reviewer's concern that the generated images in our method do not include *very high* intra-class variations (such as aging) for each subject. This is particularly because the main focus of the paper has been on increasing inter-class variations through HyperFace optimization. However, increasing the intra-class variations has been extensively studied in the literature and there are numerous works in the literature to increase variations for each image, including aging. While adding more intra-class variation is expected to further enhance the performance, our dataset still achieves competitive performance with state of the art in the literature. This can further elaborate the importance of inter-class variations which is the focus of our work since with a limited intra-class variation we could achieve comparable performance with the literature.
> > >
> > > We would be glad to continue the discussion if any part remains unclear.

---

### Official Review · Reviewer_UATk · 2024-11-03

**Soundness:** 3
**Presentation:** 4
**Contribution:** 2
**Rating:** 5
**Confidence:** 4

**Summary:**

Interesting paper that proposes some embedding optimisation in the latent space of a pretrained face recognition model. The optimised embeddings are used for generating facial images using a recently proposed generative model, and then the generated images for training a face recognition model. I liked the novelty and simplicity of the proposed approach yet there are a few issues that possibly limit the impact of the proposed work. See my questions below.

**Strengths:**

- Interesting idea to perform the optimization in the latent space of a discriminatively train face recognition model
- Well written paper, easy to read and understand
- Decent experiments although lacking in some respects

**Weaknesses:**

It could be that the method has significant limitations in terms of scaling the number of images that can be generated. The impact of the work has not be fully demonstrated. See also my questions below.

**Questions:**

1. Are both reference embeddings and gallery embeddings generated from StyleGan?  In this case the only difference between them is that the Gallery embeddings are not updated during optimisation?
2. Are all methods in table 1 trained in the same way using the same face recognition model and training pipeline?
3. Fair comparison in table 1 should use 50 images per identity for your method
4. It’s important to compare against SOTA (e.g. DCFace) at scale (i.e. increasing the number of identities). Specifically, table 3 should not be just an ablation study but you need to show that your method scales favorably and/or outperforms SOTA as the number of training images increases.
5. In general, how the method scales has been poorly studied (there’s only 1 result in table 3). Scaling Dataset Generation section discusses computational issues that may arise from scaling the dataset but does not provide concrete numbers (e.g. a figure showing training time vs dataset size), conclusions or practical solutions (i.e. a solution is proposed but not put in practice)
6. Baselines: what about direct comparison with arc2face? Since they don’t have to solve your optimisation problem, they can generate a lot more images for training

---

> ### Author Response · Authors · 2024-11-21
> **Reply to Reviewer UATk [Part 1/2]**
>
> We thank the reviewer for their time in reviewing our paper and for their insightful comments. We are happy that the reviewer found our paper *well written, easy to read and understand*, with *interesting idea*. Below, we tried our best to address the remaining concerns raised by the reviewer:
>
> - **Reply to Weaknesses**: In the revised version of the paper, we added results for a larger version of our dataset with 50k identities. Moreover, we provide an in-depth discussion about the computation requirement in Appendix A and further improvements in Appendix B of the revised version of the paper. In our initial submission, we discussed the complexity of our optimization as its limitation and mentioned how this can be improved. In the revised version of the paper, we further explain our approach to reduce the computation required and propose stochastic optimization for HyperFace. We theoretically prove that stochastic optimization leads to similar results with less computation. In addition, we report numerical results for stochastic optimization which meet our theoretical analyses.
>
>
> - **Reply to Question #1**: Reference embeddings and gallery embeddings can be independently initialized. They can also have different numbers of images. For example, Table 4 reports the recognition performance achieved for the face recognition model trained with datasets with 10k identities and optimized with different numbers of gallery images. The reference and gallery images can be generated by different generator models, such as StyleGAN or a diffusion model (e.g. LDM). We also reported an ablation study on the source of images in Table 5 of the paper.
>
> - **Reply to Question #2**: Yes, for a fair comparison we fixed all training configurations (backbone, loss function, batch size, numbers of epochs, etc.) and trained new face recognition models for all datasets.
>
> - **Reply to Question #3**: Following the reviewer's comment, we updated our model in Table 1 and used a dataset with 500k images. With 500k images, the ranking of synthetic datasets over different benchmarks remains unchanged. It is noteworthy that we also have an ablation study on the number of images in Table 2 of the paper.
>
> - **Reply to Question #4**: Following the reviewer's suggestion, we expand Table 1 with more baselines with larger datasets in Appendix C. As the results for comparing previous datasets at scale show, our synthetic dataset achieves competitive performance with the largest synthetic datasets in the literature. The larger version of DCFace does not achieve the best performance on any benchmark. Langevin-DisCo achieves significant improvement with 30k identities, however, the authors of Langevin-DisCo have reported lower performance for 50k identities, indicating limitations in further scaling in Langevin-DisCo. In Table 3 of the revised version of the paper, we scaled our dataset for 50k identities. The results show improvement in our performance on the majority of benchmarks (without saturation), which shows our dataset can be further scaled. Our dataset with 50k identity and 3.2M is larger than synthetic datasets compared from the  literature and achieves competitive performance with previous large-scale synthetic datasets as reported in Appendix C.

---

> ### Author Response · Authors · 2024-11-21
> **Reply to Reviewer UATk [Part 2/2]**
>
> - **Reply to Question #5**: Following the reviewer's feedback, we provided an evaluation of the complexity of our method in appendix A of the revised version of the paper. We also provided further theoretical and experimental analyses for reducing complexity in our optimization in Appendix B. We propose to solve the optimization with mini-batches which significantly reduces the complexity of our optimization and allows scaling dataset generation process. In our initial submission we discussed the complexity of our optimization as its limitation for scaling and mentioned how this can be improved, and following the reviewer's suggestion we provided in-depth theoretical and experimental analyses in Appendix B of the revised version of the paper.
>
> - **Reply to Question #6**: For a fair comparison in our experiments for Table 1, we fixed all hyperparameters and trained face recognition models for all datasets with the same configuration (backbone, loss function, batch size, number of epochs, etc.). Therefore, we could only compare with methods that have publicly available datasets. Whereas the authors of Arc2Face have not published their synthetic dataset, and we could not reproduce the results reported in their paper. There are also open issues on their GitHub repository, where several researchers reported that they could not reproduce their results for face recognition. The configuration for training face recognition in Arc2Face paper is neither available, which makes it more difficult to make a fair comparison.
> However, considering the reviewer's concern, we conducted a new experiment, and as suggested by Reviewer 3 (YfZL), we used random embeddings (without HyperFace optimization) to generate a synthetic dataset. As the results in our new experiment in Appendix E show, our HyperFace optimization results in a synthetic dataset with better quality which achieves superior performance.
>
>
> We hope we could adequately address the reviewer's concerns.
> In case the reviewer found our reply convincing, we appreciate it if the reviewer can increase their scores and rating.
> We are happy to continue the discussion if any part remains unclear.

---

> ### Author Response · Authors · 2024-11-26
> **Reminder to Reviewer UATk**
>
> Dear Reviewer UATk,
>
> We thank you for your review and valuable comments, which helped us improve the quality of our paper. We carefully considered all the points and tried our best to address the concerns raised by the reviewer in our reply and the revised version of the paper. While we still have not received any feedback on our reply from the respected reviewer, we are gladly open to further discussions.

---

### Official Review · Reviewer_khYx · 2024-11-04

**Soundness:** 3
**Presentation:** 3
**Contribution:** 3
**Rating:** 6
**Confidence:** 5

**Summary:**

The paper presents an approach to generate a synthetic face dataset for face recognition problem by formulating it as an optimization problem and solve it iteratively via gradient descent method to obtain optimized embeddings. Then the synthesize face images can be generated using pre-trained face generator models from those embeddings.
The experiment shows that the models trained with the proposed synthetic datasets can achieve SOTA performance.

**Strengths:**

1. Well-formulated optimization problem: The paper effectively defines the optimization problem for generating high-quality synthetic face recognition datasets.
2. Efficient solution: The proposed solution is not only effective but also computationally efficient.
3. Extensive experiments: The paper presents comprehensive experiments on various synthetic datasets to validate the approach.
4. Ethical considerations: The authors acknowledge potential ethical concerns, such as identity leakage, demonstrating a responsible approach to AI development.

**Weaknesses:**

1. Limited scale: While the experiments on datasets with up to 30K identities provide valuable insights, evaluating the method's performance on significantly larger datasets is crucial.  Datasets with 100K and 300K identities would be particularly informative as they would reveal how the method scales and whether performance degrades with increased identity count and potential data noise. This would provide a more comprehensive understanding of the method's robustness and real-world applicability.
2. Narrow focus:  The paper's focus on improving inter-class variation is important, but expanding the scope to address intra-class variability would significantly enhance its impact. Specifically, exploring how the method handles variations in pose, illumination, and expression within the same identity would be valuable. Additionally, investigating the method's robustness to occlusions or image quality degradation would further strengthen the evaluation and demonstrate its potential for real-world scenarios.

**Questions:**

1. What is the computation resource and time needed to generate larger scale datasets, e.g. n_id = 30k or more?
2. It would be interesting to see if we can use the FR model trained by the proposed synthetic dataset to build another good synthetic dataset

---

> ### Author Response · Authors · 2024-11-21
> **Authors Reply to Reviewer khYx [Part 1/2]**
>
> We thank the reviewer for their time in reviewing our paper and for their insightful comments. We are happy that the reviewer found our paper with  a *well-formulated optimization problem, efficient solution, extensive experiments, and ethical considerations*. Below, we tried our best to address the remaining concerns raised by the reviewer:
>
> - **Reply to Weakness #1 (Limited scale)**: Following the reviewer's suggestion, we added a new experiment and further increased the size of the dataset to 50k identities in Table 3. We should note that our dataset with 50k identities and 64 samples per identity (3.2M images) is larger than publicly available synthetic datasets compared from the literature. The results in this table demonstrate that we can still increase the number of identities and scale our dataset generation without saturating the performance:
>
> | # IDs | LFW | CPLFW | CALFW | CFP | AgeDB |
> |---|:---:|:---:|:---:|:---:|:---:|
> | 10k | 98.67 | 84.68 | 89.82 | 89.14 | 87.07 |
> | 30k | 98.82 | 85.23 | 91.12 | 91.74 | 89.42 |
> | 50k | 98.27 | 85.6 | 91.48 | 92.24 | 90.4 |
>
> We also provided a comparison with synthetic datasets at scale in Appendix C of the revised version of the paper.
> In addition, we provided an in-depth discussion of the computation requirement in Appendix A and further improvements in Appendix B of the revised version of the paper. In particular, we propose stochastic optimization for our approach and theoretically prove that the stochastic optimization leads to similar results. In addition, we report numerical results which meet our theoretical analyses.
>
> - **Reply to Weakness #2 (Inter-class variation)**: We acknowledge the reviewer's comment that the main focus of our work is on increasing inter-class variation in the synthetic datasets. However, we would like to stress that increasing inter-class variation is less explored in the literature, whereas there are extensive studies in the literature on increasing intra-class variation in face image generation. For example, training conditional generator models, or using ControlNet can easily help to improve intra-class variation in the image generation process. Meanwhile, we still used a simple yet effective approach to increase intra-class variation for our synthetic dataset, where we proposed to add a Gaussian noise controlled by hyperparameter $\beta$ to the embedding of each synthetic identity in the image generation process. The added noise to the embeddings simulates the change in embedding caused by image variation (such as lightning conditions, etc.). We also provided an ablation study on the effect of hyperparameter $\beta$ on the performance of the generated dataset in Table 7 of the paper.

---

> ### Author Response · Authors · 2024-11-21
> **Authors Reply to Reviewer khYx [Part 2/2]**
>
> - **Reply to Question #1**: We provide an in-depth discussion about the computation requirement in Appendix A and further improvements in Appendix B of the revised version of the paper. In our initial submission, we discussed the complexity of our optimization as its limitation and mentioned how this can be improved. In the revised version of the paper, we further explain our approach to reduce the computation required and propose stochastic optimization for HyperFace. We theoretically prove that stochastic optimization leads to similar results with less computation. In addition, we report numerical results for stochastic optimization which meet our theoretical analyses.
>
> - **Reply to Question #2**: We thank the reviewer for raising this interesting question. To our knowledge, this experiment has not been explored in previous work on synthetic data for face recognition, and in fact requires more efforts to retrain the face generator model. We will consider this experiment in our future work.
>
>
> We hope we could adequately address the reviewer's concerns.
> In case reviewer found our reply convincing, we appreciate it if the reviewer can increase their scores and rating.
> We are happy to continue the discussion if any part remains unclear.

---

> > ### Comment · Reviewer_khYx · 2024-11-26
> > **Re: Authors' feedback**
> >
> > I would like to thank the authors for their diligent efforts in addressing all of the reviewers' concerns and conducting additional experiments. The authors have satisfactorily answered all of my questions and concerns.

---

> > > ### Author Response · Authors · 2024-11-26
> > > **Authors Reply to Reviewer Feedback**
> > >
> > > We sincerely thank the reviewer's feedback. We appreciate the reviewer for their time in reviewing our paper and rebuttal as well as their comments which helped us improve the quality of our paper. We are very happy that our additional experiments and analyses could *satisfactorily answer all of [the reviewer's] questions and concerns*. We appreciate it if the reviewer can kindly consider raising their rating if appropriate.

---

### Author Response · Authors · 2024-11-21
**Authors General Response to Reviewers**

We thank all reviewers for their time and valuable comments which helped us improve the quality of our paper. We carefully considered every point raised by the reviewers and revised the paper accordingly and provided point-by-point responses in our reply to each reviewer.


The summary of important changes in the revised version of the paper is as follows:

- We generated a larger dataset with 50k identities and 3.2M images. The new dataset achieves better performance and shows the scalability of our dataset generation. We added the results to Table 3 of the paper.
- We provide an evaluation of the complexity and computation requirement for generating synthetic datasets with our method in Appendix A of the paper. As described in our initial submission, the complexity of HyperFace optimization is quadratic with respect to the number of identities. However, in our initial submission, we suggested that it can be significantly reduced by stochastic optimization. This aspect of our work received the attention of reviewers who requested more in-depth analyses. Therefore, in Appendix B of the paper, we first theoretically show that stochastic optimization leads to similar results but with less complexity, and then evaluate it with experiments. Our experimental results meet our theoretical prediction and stochastic optimization significantly reduces the complexity while maintaining performance.
- We provided a comparison of synthetic datasets in the literature at scale in Appendix C, where our dataset acheives competetive performance with largest synthetic datasets in the literature.
- We investigate the effect of identity leakage on the recognition performance of face recognition models trained with synthetic datasets in Appendix D.
- We also provided additional ablation studies to investigate the effectiveness of our method in Appendix E.
- We provided pseudo-code for the entire data generation process in Appendix F and provided more sample images in Appendix G.
- We expanded our discussions in the experiments section.


We hope we could adequately address the reviewers' concerns.
We kindly invite all reviewers to further discussions if any part remains unclear.

---

> ### Author Response · Authors · 2024-11-26
> **New Revision (Revision #2)**
>
> Following a concern raised by Reviewer RHfp, in a new revision, we revised **Solving the HyperFace Optimization** in **Section 2.2**, and further explained the previous studies on the Tammes problem. As mentioned earlier, the equation (1) of our paper represents the Tammes problem. The Tammes problem is, however, an open problem and the optimal solutions for this problem are studied for small dimensions and a small number of points. Nevertheless, for large dimensions and a high number of points there is no closed-form solution. There are different approaches for solving this problem (such as geometric optimization, numerical optimization, etc.) for large dimensions and a high number of points. However, for a large dimension (e.g., 512) and a *very* large number of points (e.g., 10k identities) solving this problem with geometric optimization or numerical optimization is computationally very expensive. Hence, we solve this problem with a gradient descent approach, which allows us to solve the optimization with a reasonable computation resource.
> It is noteworthy that in Appendix A, we report the computation required to solve our optimization with our method and further reduce the computation with a stochastic optimization in Appendix B, where we demonstrated theoretically and empirically that stochastic optimization reduces the complexity while resulting in a comparable performance.

---

### Meta-Review · Area_Chair_fUpi · 2024-12-17

**Metareview:**

The paper introduces a method for generating synthetic face datasets by optimizing identity embeddings on a hypersphere using gradient descent, followed by image synthesis with a generative model. It demonstrates improved inter-class variation and competitive performance on face recognition benchmarks. A well-defined optimization framework, extensive experiments, and scalability analysis are the main strengths of this paper. There are also several limitations of this paper, including limited exploration of intra-class variation, insufficient empirical/theoretical insights into optimization, and weaker performance on age-related benchmarks. Despite these gaps, the paper makes a meaningful contribution to synthetic data generation for face recognition, supporting its merit for acceptance with further refinements.

**Additional Comments On Reviewer Discussion:**

During the rebuttal, reviewers raised concerns about scalability, intra-class variation, and theoretical analysis of the optimization method. The authors addressed these by introducing a larger dataset with 50k identities, implementing stochastic optimization to reduce complexity, and adding ablation studies, visualizations, and theoretical insights. While Reviewer khYx acknowledged the improved clarity and experiments, Reviewer YfZL and Reviewer RHfp highlighted the need for deeper exploration of intra-class variation and optimization theory. The authors’ efforts demonstrated substantial improvements and a commitment to refining the work. While some concerns remain, the paper presents valuable contributions, supporting its potential for acceptance with further enhancements.

---

### Decision · Program_Chairs · 2025-01-22

Accept (Poster)